# Compelling ReLU Networks to Exhibit Exponentially Many Linear Regions at Initialization and During Training

**Max Milkert** [1 2]   **David Hyde** [1]   **Forrest Laine** [1]

## Abstract

In a neural network with ReLU activations, the number of piecewise linear regions in the output can grow exponentially with depth. However, this is highly unlikely to happen when the initial parameters are sampled randomly, which therefore often leads to the use of networks that are unnecessarily large. To address this problem, we introduce a novel parameterization of the network that restricts its weights so that a depth $d$ network produces exactly $2^d$ linear regions at initialization and maintains those regions throughout training under the parameterization. This approach allows us to learn approximations of convex, one-dimensional functions that are several orders of magnitude more accurate than their randomly initialized counterparts. We further demonstrate a preliminary extension of our construction to multidimensional and non-convex functions, allowing the technique to replace traditional dense layers in various architectures.

## 1. Introduction

Beyond complementary advances in areas like hardware, storage, and networking, the success of neural networks is primarily due to their ability to efficiently capture and represent nonlinear functions (Gibou et al., 2019). In a neural network, the goal of an activation function is to introduce nonlinearity between the network's layers so that the network does not simplify to a single linear function. The rectified linear unit (ReLU) has a unique interpretation in this regard. Since it either deactivates a neuron or acts as an identity, the resulting transformation on each individual input remains linear. However, each possible configuration of active and inactive neurons can produce a unique linear transformation over a particular region of input space. The number of these activation patterns and their corresponding linear regions provides a way to measure the expressivity of a ReLU network[1] and can theoretically scale exponentially with the depth of the network (Montufar et al., 2014; Serra et al., 2018). Hence deep architectures may outperform shallow ones.

Surprisingly, though, a sophisticated theory of how to best encode functions into ReLU networks is lacking, and in practice, adding depth is often observed to help less than one might expect from this exponential intuition. Lacking more advanced theory, practitioners typically use random parameter initialization and gradient descent, the drawbacks of which often lead to extremely inefficient solutions. Indeed, Hanin & Rolnick (2019) show a rather disappointing bound for randomly initialized networks: the average number of linear regions formed at initialization is invariant with depth. Whether using a deep or shallow architecture, only the total number of neurons matters. They further observed that gradient descent has a difficult time creating new activation regions, and thus their bounds approximately held after training. As we will discuss later, the total number of linear regions is not a "local" property in parameter space that gradient descent can directly optimize (see Figure 5). Gradient descent is also prone to redundancy; for instance, Frankle & Carbin (2019) show how around 95% of weights may ultimately be eliminated from a network without significantly degrading accuracy.

The present work presents mathematical algorithms that avoid the limitations of random initialization. Our contributions include:

- A reparameterization of a 4-neuron-wide, depth $d$ ReLU network that constrains it to initialize with $2^d$ activation regions over the input domain (Section 3)

- A novel pretraining strategy, which enforces the existence of $2^d$ activation regions during optimization to produce its initial solution (Sections 3, 3.1)

[1]Department of Computer Science, Vanderbilt University, Nashville TN, USA [2]National Renewable Energy Laboratory, Golden CO, USA. Correspondence to: Max Milkert <max.milkert@vanderbilt.edu>.

*Proceedings of the $42^{nd}$ International Conference on Machine Learning*, Vancouver, Canada. PMLR 267, 2025. Copyright 2025 by the author(s).

---

[1]See Appendix A.2 for definitions of terms like linear regions, activation patterns, and activation regions.

- Numerical results for one-dimensional test cases that yield orders-of-magnitude improvements in network performance (Section 4)

- Extensions to non-convex and multidimensional functions, and use of our method as a drop-in replacement for dense layers in arbitrary networks (Sections 5 and 6; Appendices B.2–B.3)

While an exponential increase in the expressiveness of a ReLU network does not necessarily imply an exponential increase in performance, one may intuitively expect a substantial benefit, and our results bear this out, yielding error values orders of magnitude lower than a traditionally-trained network of equal size for simple test problems (Section 4), and outpacing such networks' performance during pretraining on more practical test cases such as ImageNet (Section 6). We emphasize that while the preliminary mathematical analysis in this work pertains to a specific ReLU construction, this construction can be repeated and combined in useful ways to obtain performance improvements on larger networks (Section 5). We conclude in Section 7 with discussions on extending the mathematical framework of our method to arbitrary ReLU networks and other types of networks, which would have significant practical utility.

## 2. Related Work

This work is primarily concerned with a novel training methodology for ReLU networks, but its development stems from a more abstract idea of an exponentially accurate approximation to the function $x^2$, which can be encoded into a ReLU network. Our work generalizes this construction into a trainable family of differentiable convex one-dimensional functions (see Appendix A.3).

### 2.1. Function Approximation

Infinitely wide neural networks are known to be universal function approximators, even with only one hidden layer (Hornik et al., 1989; Cybenko, 1989). Infinitely deep networks of fixed width are universal approximators as well (Lu et al., 2017; Hanin, 2019). In finite cases, one may study trade-offs between width and depth to assess a network's ability to approximate (learn) a function.

Notably, there exist functions that can be represented with a sub-exponential number of neurons in a *deep* architecture, yet which require an exponential number of neurons in a *wide and shallow* architecture. For example, Telgarsky (2015) shows that deep neural networks with ReLU activations on a one-dimensional input are able to generate symmetric triangle waves with an exponential number of linear segments (shown in Figure 1 as the ReLU network $T(x)$). This network functions as follows: each layer

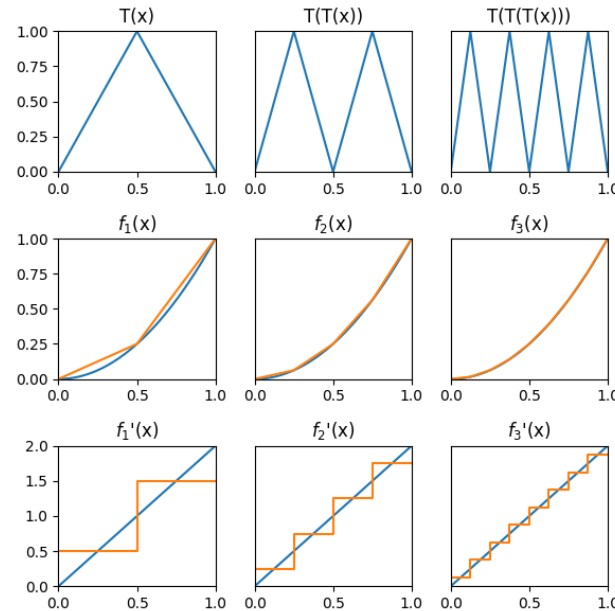

*Figure 1.* (Top to bottom) Composed triangle waves; using collections of the above function to approximate $x^2$; derivatives of the above approximations.

takes a one-dimensional input on $[0, 1]$, and outputs a one-dimensional signal also on $[0, 1]$. The function they produce in isolation is a single symmetric triangle. Together in a network, each layer inputs its output to the next, performing function composition. Since each layer converts lines from 0 to 1 into triangles, it doubles the number of linear segments in its input signal, exponentially scaling with depth.

The same effect can be achieved with non-symmetric triangle waves (Huchette et al., 2023) (or any shape that sufficiently "folds" the input space (Montufar et al., 2014)). Our reparameterization strategy (Section 3) focuses on non-symmetric triangle waves.

The dilated triangular waveforms produced in this manner are not particularly useful on their own. Their oscillations quickly become excessively rapid, and their derivatives do not exist everywhere (especially in the infinite-depth limit). But these problems can be rectified by taking a sum over the layers of a network. Yarotsky (2017) and Liang & Srikant (2016) construct $y = x^2$ on $[0, 1]$ with exponential accuracy using symmetric triangle waves. To produce their approximation, one begins with $f_0(x) = x$, then computes $f_1(x) = f_0(x) - T(x)/4$, $f_2(x) = f_1(x) - T(T(x))/16$, $f_3(x) = f_2(x) - T(T(T(x)))/64$, and so forth, as pictured in Figure 1. As these successive approximations are computed, Figure 1 plots their convergence to $x^2$, as well as the convergence of the derivative to $2x$. Our reparameterization generalizes this approximation to use non-symmetric

triangle waves to approximate a wider class of convex, differentiable, one-dimensional functions.

The $x^2$ approximation is used by other theoretical works as a building block to guarantee exponential convergence rates in more complex systems. Perekrestenko et al. (2018) construct a multiplication gate via the identity $(x + y)^2 = x^2 + y^2 + 2xy$. The squared terms can all be moved to one side, expressing the product $xy$ as a linear combination of squared terms. They then further assemble these multiplication gates into an interpolating polynomial, which can have an exponentially decreasing error when the interpolation points are chosen to be the Chebyshev nodes. Polynomial interpolation does not scale well into high dimensions, so this and papers with similar approaches will usually come with restrictions that limit function complexity: Wang et al. (2018) requires low input dimension, Montanelli et al. (2020) uses band limiting, and Chen et al. (2019) approximates low-dimensional manifolds. These works all make use of a fixed representation of $x^2$. If our networks were substituted in for the $x^2$ approximation, these works would provide theoretical guarantees about the capabilities of the resulting model. Even though the approximation rates will not scale well with input dimension, they serve as a bound that can be improved upon. In Section 5, we further elaborate on how to use our networks to represent higher-dimensional or non-convex functions.

Other works focus on showing how ReLU networks can encode and subsequently surpass traditional approximation methods (Lu et al., 2021; Daubechies et al., 2022), including spline-type methods (Eckle & Schmidt-Hieber, 2019). Interestingly, certain fundamental themes from above like composition, triangles, or squaring are still present. Another interesting comparison of the present work is to Ivanova & Kubat (1995), which uses decision trees as a means to initialize sigmoid neural networks for classification. Similar to the spirit of our work, which restricts parameterizations of ReLU networks, Elbrächter et al. (2019) explores theoretical aspects of the conditioning of ReLU network training and provides constructive results for a parameterization space that is well-conditioned. Chen & Ge (2024) present a creative approach where they explore reparameterizing the direction of weight vectors using hyperspherical coordinates to improve training dynamics. Unlike their reparameterization, ours will restrict the network's expressivity in order to prevent it from learning inefficient weight patterns. Lastly, Park et al. (2021) approaches the problem of linear region maximization from an information theory perspective and uses a loss penalty rather than a reparameterization to increase the number of linear regions.

## 2.2. Neural Network Initialization

Our work seeks to improve network initialization by making use of explicit theoretical constructs. This stands in sharp contrast current standard approaches, which treat neurons homogeneously. Two popular initialization methods implemented in PyTorch are the Kaiming (He et al., 2015) and Xavier initialization (Glorot & Bengio, 2010). They use weight values that are randomly sampled from distributions defined by the input and output dimension of each layer. Aside from sub-optimal approximation power associated with random weights, a common issue is that the initial weights and biases in a ReLU network can cause every neuron in a particular layer to output a negative value. The ReLU activation then sets the output of that layer to 0, blocking any gradient updates. This is referred to as the dying ReLU phenomenon (Qi et al., 2024; Nag et al., 2023). Worryingly, as depth goes to infinity, the dying ReLU phenomenon becomes increasingly likely (Lu et al., 2020). Several papers propose solutions: Shin & Karniadakis (2020) use a data-dependent initialization, while Singh & Sreejith (2021) introduce an alternate weight distribution called RAAI that can reduce the likelihood of the issue and increase training speed. We observed during our experiments that RAAI greatly reduces, but does not eliminate, the likelihood of dying ReLU. Our approach enforces a specific network structure that does not collapse in this manner.

## 3. Initialization and Pretraining Construction

In this section we describe our ReLU network reparameterization strategy that ensures an exponential number of linear regions with respect to depth. We focus on architecting the weights of a 4-neuron-wide, arbitrary depth ReLU network; while this is a constraint of the present manuscript, we will show that despite this, we can effectively apply our construction to a wide variety of examples.

As alluded to in Section 2.1, the overall strategy of our reparameterization and initialization algorithm is to associate a nonsymmetric triangle function with each layer of the network. This yields a reparameterization of the ReLU network in terms of the locations of each triangle's peak on $(0, 1)$, $a_i$, rather than in terms of the network's weights. Unlike the raw weights of a depth $d$ ReLU network, *any* values chosen for the triangle peak parameters will result in the creation of $2^d$ linear regions. Furthermore, this parameterization is trainable. By using the peak location to set the raw weights of a layer, the gradients can backpropagate through the raw weights to update the triangle peaks, effectively confining the network to a subspace of weights that generates many linear regions. Pseudocode for our entire algorithm is provided in Appendix A.1.

To illustrate how we set weights, we first describe the math-

emetical functions that arise in our analysis. Triangle functions are defined as

$$T_i(x) = \begin{cases} \frac{x}{a_i} & 0 \le x \le a_i, \\ 1 - \frac{x-a_i}{1-a_i} & a_i \le x \le 1 \end{cases},$$

where $0 < a_i < 1$. This produces a triangular shape with a peak at $x = a_i$ and zeros at $x = 0$ and $x = 1$. These functions are implemented by two ReLU neurons in each layer. In a deep network, each layer composes its triangle function with the output of the layer before it. The result is that each layer computes the following triangular waveform:

$$W_i(x) = \bigcirc_{j=0}^{i} T_j(x) = T_i(T_{i-1}(...T_0(x))). \quad (1)$$

These triangle waves will have $2^i$ linear regions, doubling with each layer. By allocating an extra neuron in each layer, our construction will take a weighted sum over each layer's triangle wave, using them as a deep basis to build the output. In the infinite depth limit, the network output takes the form:

$$F(x) = \sum_{i=0}^{\infty} s_i W_i(x), \quad (2)$$

where $s_i$ are scaling coefficients on each of the composed triangular waveforms $W_i$. $F(x)$ would be a parabola if we exactly followed the related work. $F(x)$ could also end up being a fractal curve if the $s_i$ are chosen to be too large. Choosing $s_i$ carefully to regularize $F(x)$ is a major part of our construction.

Now we will discuss how to encode these functions into the weights of a ReLU network, beginning with a network for a single triangle function (see the upper left diagram in Figure 2). Each triangle function has two nonzero linear pieces, requiring two ReLU neurons. For simplicity, we will set both neurons' input weights to 1. We will arrange for neuron $t_1$ to be active for all inputs, and since the maximum output should be 1 at the peak location $a \in (0, 1)$, neuron $t_1$ will have an outgoing weight of $1/a$. $x = a$ is where the slope should change, so neuron $t_2$ will be biased by $-a$ to suppress it for $x < a$. Since both neurons are active for $x > a$, neuron $t_2$ will need to be weighted by $-(1/a + 1/(1-a)) = -1/(a-a^2)$; this weight will both suppress neuron $t_1$ and also ensure the second leg of the triangle function reaches 0 at $x = 1$. This is now the first network shown in Figure 2.

To compose the triangle waves $W_i(x)$, copies of this network can be stacked together depth-wise, lining up the "in" and "out" neurons of successive networks. This will have a non-uniform width ($1 \to 2 \to 1 \to 2 \to 1...$), but since the input weights of the hidden $t_1$ and $t_2$ neurons are both 1, it can be simplified into the form shown on the top right of Figure 2. The weights $1/a$ and $-1/(a-a^2)$ give the correct

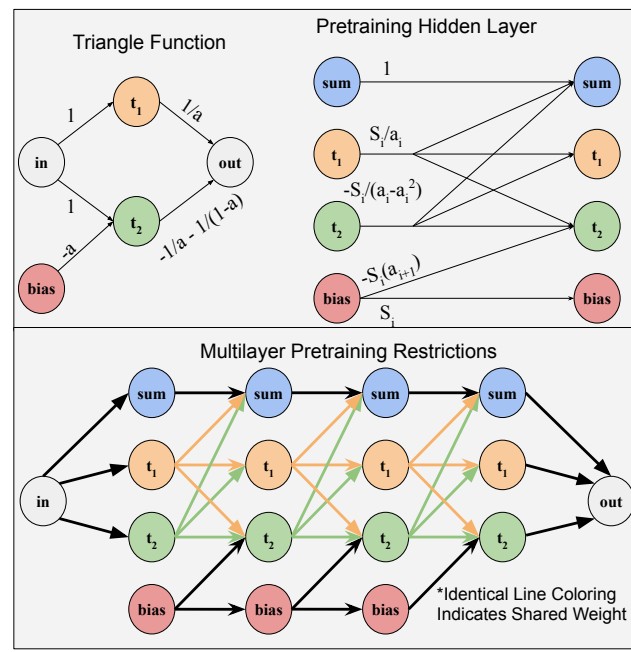

*Figure 2.* On the top left is a network representation of a triangle function. The top right shows that triangle function as a hidden layer of a network. The one-dimensional input and output of a triangle function is converted into shared weights. A full pretraining network is assembled on the bottom.

way to combine the $t_1$ and $t_2$ neurons in order to obtain a triangle function, so every hidden neuron will combine the previous $t_1$ and $t_2$ neuron in this proportion. Using identical weights makes each layer act as if it is composing functions in one dimension, rather than in a dimension equal to the number of neurons.

The goal of the network is not to output triangle waves; instead, the function we seek to approximate is $F(x)$, defined earlier to be a weighted sum of the triangle waves from each layer. We add an extra neuron to each layer, labeled "sum" in Figure 2, to compute this. These are ordinary neurons, but thanks to their specific weights, they will act similarly to a residual connection (He et al., 2016), allowing the hidden features in the network to be used in the output. Each sum neuron adds the previous layer's triangle wave to the previous sum neuron, thereby iteratively updating an approximation to $F(x)$.

According to Theorem 3.1, each triangle wave added to the sum must have an amplitude exponentially smaller than the last. Exponentially small weights would pose conditioning issues in optimization, so rather than giving the sum neuron an exponentially small weight, we use the ratio of successive scaling coefficients $S_i = s_i/s_{i-1}$ to iteratively decay the amplitude of the $t_1$ and $t_2$ neurons. But because $t_2$ now has

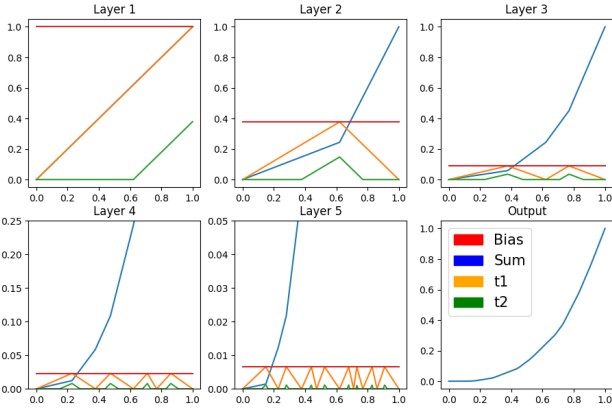

*Figure 3.* Each colored line shows the output signal of a neuron with respect to the input to the network. Colors match the corresponding neurons in Figure 2.

an exponentially small amplitude, its bias must be exponentially small to function properly. Rather than implementing an exponentially small bias, we make the bias into a neuron. This is rather unconventional, but by using the connection to the previous bias neuron, the bias can decay iteratively, and $t_2$ neurons can use their weight to the bias neuron to implement their bias. In matrix form, our hidden layers are:

$$\begin{bmatrix} 1 & \pm[S_i/a_i & -S_i/(a_i - a_i^2)] & 0 \\ 0 & S_i/a_i & -S_i/(a_i - a_i^2) & 0 \\ 0 & S_i/a_i & -S_i/(a_i - a_i^2) & -S_i a_{i+1} \\ 0 & 0 & 0 & S_i \end{bmatrix} \times \begin{bmatrix} \text{sum} \\ t_1 \\ t_2 \\ \text{bias} \end{bmatrix}.$$

Here, rows correspond to neurons, and the sum neuron can either add or subtract the triangle waves depending on whether the target function is convex or concave.

### 3.1. Pretraining and Overall Algorithm

We imagine the process of optimizing neural network parameters as three phases: (1) reparameterization and initialization, where a network's weights are initially set; (2) pretraining, where the network is trained under that reparameterization; and (3) training, where the network is trained using the network's weights directly. Our pretraining algorithm acts as a preconditioner for or guide to the loss landscape, and when an optimizer is close to a minimum, the exponential expressivity of the network may no longer need to be guaranteed to ensure efficient neuron usage.

Our pretraining algorithm is thus to simply use an optimizer like gradient descent to train the network under the reparameterization described above. Explicit formulas for updating the reparameterized weights, and pseudocode for our whole algorithm, are listed in Appendix A.1.

### 3.2. Differentiable Model Output

Given the rapid oscillations of the triangle waves formed at each layer, the network will output a fractal with many choices of scaling parameters. This would be poorly predictive of unseen data points generated by a smooth curve. Rather than turning to a general regularizer like batch norm (Ioffe & Szegedy, 2015) or layer norm (Lei Ba et al., 2016), we use a regularization scheme that arises from the form of our functions. When we require that the network output approach a differentiable function in the infinite depth limit, an elegant relationship arises where the peak locations of the triangle waves uniquely determine the scales with which to sum them.

**Theorem 3.1.** $F'(x) = \sum_{i=0}^{\infty} s_i W_i'(x)$ *and is continuous on* $[0, 1]$ *only if the scaling coefficients* $s_i$ *are selected based on the triangle peaks* $a_i$ *according to:*

$$s_{i+1} = s_i(1 - a_{i+1})a_{i+2}. \tag{3}$$

Mathematical analysis and sufficient conditions for differentiability in the limit are in Appendix A.3. In Section 4, we will see that following this regularization allows the pretraining phase to produce better initial solutions.

## 4. One-Dimensional Experiments

Since our network construction applies directly to one-dimensional convex functions, we focus on such functions in this section (we cover more difficult functions in Sections 5–6 and Appendices B.2–B.3). The aim of these experiments is twofold: (1) we would like to determine the most effective function representations possible from such a small network, and (2) to explore how the utilization of an increased number of linear regions can affect a network's ability to capture underlying nonlinearity in its training data. To demonstrate that our networks can learn function representations that better utilize depth, we benchmark against PyTorch's (Paszke et al., 2019) default settings (nn.linear() uses Kaiming initialization), as well as the RAAI distribution (Singh & Sreejith, 2021), and produce errors that are *orders of magnitude lower than both*. We also train several permutations of our reparameterized networks, which validate that our pretraining and our differentibility constraints indeed facilitate smoother navigation of the loss landscape. Lastly, we conduct a second round of tests to determine if pretrained networks display an enhanced predictive capacity on unseen data points, as might be expected if they can leverage greater nonlinearity in their outputs.

### 4.1. Experimental Setup

All models are trained using Adam (Kingma & Ba, 2017) as the optimizer with a learning rate of 0.001 for 1,000 epochs

to ensure convergence (see Appendix B.1 for a brief analysis of different learning rates). Each network is four neurons wide with five hidden layers, along with a one-dimensional input and output. The loss function used is the mean squared error, and the average and minimum loss are recorded for 30 models of each type. To compare how well each setup navigates the loss landscape, the networks using triangle-based parameterizations share a common set of starting locations where the triangle peaks and scaling parameters are chosen according to Theorem 3.1. The "unregularized" network has extra degrees of freedom to choose the scaling coefficients after initialization (so that differentiability of the output is not enforced during pretraining). "pretraining skipped" is a network that initializes with our method, but only trains in the standard parameterization. The four curves we approximate are $x^3$, $x^{11}$, $\tanh(3x)$, and a quarter period of a sine wave. To approximate the sine and the hyperbolic tangent, the triangle waves are added to the line $y = x$, while for $x^3$ and $x^{11}$ the waves are subtracted. When the waves are subtracted, the first scaling factor has to be changed to $a_0 * a_1$ instead of $(1 - a_0) * a_1$. The first set of data is 500 evenly spaced points on the interval $[0, 1]$ for each of the curves. This is chosen to be very dense deliberately, to try to evoke the most accurate function interpolations from each network. We determine from these tests that pretraining with differentiability enforced produces the best results, so we compare it to standard networks in our second set of experiments. We use a second set of data consisting of only 10 points, with a test set of 10 points spaced in between so as to be as far away from learned data as possible. This set of experiments evaluates the predictive capacities of the networks on unseen data.

### 4.2. Numerical Results

Our first set of results are shown in Tables 1 and 2, wherein we observe several important trends. First, the worst performing networks are the "default networks" that rely on randomized (Kaiming) initialization (see also Figure 5). Even the networks that forgo pretraining benefit from initializing with many activation regions. When pretraining constraints are used, they are able to steer gradient descent to the best solutions, resulting in reductions in minimum error of three orders of magnitude over default networks. Pretraining with differentiability enforced also closes the gap between the minimum and mean errors compared to other setups. This indicates that these loss landscapes are indeed the most reliable to traverse. Enforcing differentiability during pretraining can impart a bias towards smoother solutions during subsequent unassisted gradient descent.

The last trend to observe is the poor average performance of default networks. In a typical run of these experiments, around half of the default networks collapse from the dying ReLU phenomenon. RAAI is able to eliminate most, but

not all of the dying ReLU instances due to its probabilistic nature, so it, too, has high mean error. This is why we have focused on presenting the minimum error over all the trials.

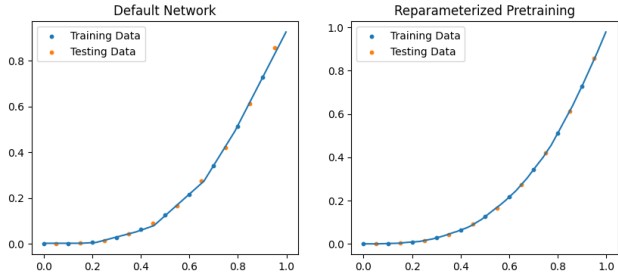

*Figure 4.* Standard Kaiming initialization/gradient descent vs. pretraining with differentiablity enforced. Using more linear regions allows the curve to better predict the test points.

Our second set of results is shown in Table 3 and in Figure 4. Here the most important impact of utilizing exponentially many linear regions is demonstrated. Not only can more accurate representations of training data be learned, but maintaining more linear regions allows the network to better capture underlying nonlinearity to enhance its predictive power in regression tasks. This result is especially significant because it indicates that even in cases where there are fewer data points than linear regions, having the additional regions can still provide performance advantages.

## 5. Extension to Non-Convex Functions and Higher Dimensions

The results presented so far are limited to one-dimensional convex functions, but our method can easily be extended. Since our constructed networks are one-dimensional, they can be used as the activation functions of a larger network (see Figure 9 in the Appendix for a schematic). Our experimental results show that even using just two copies of our network in this manner can go a long way towards increasing the expressive ability of the technique. For example, two of our networks could be used to build a two-dimensional convex function. This can be done by using one network to build a convex function oriented in the $x$ direction, and the other in the $y$ direction. A positive linear combination of their outputs then yields a two-dimensional convex function. Similarly, a non-convex function can be realized by taking a difference of the outputs of two of our networks. Of course, this pattern can be generalized to an arbitrary-width layer using $n$ of our networks.

Figures 6 and 7 show learning the non-convex function $y = x^3 - x$ and the two-dimensional convex function $z = r^3$ on $[-1, 1]$, each by using two of our constructed networks with 5 hidden layers. For both $x^3 - x$ and $r^3$, we sub-

*Table 1.* Minimum and mean (30 samples) MSE error approximating $y = x^3$ and $x^{11}$.

| Training Type | Min $x^3$ | Min $x^{11}$ | Mean $x^3$ | Mean $x^{11}$ |
|---|---|---|---|---|
| Default Network (Kaiming) | $2.11 \times 10^{-5}$ | $2.19 \times 10^{-5}$ | $7.20 \times 10^{-2}$ | $2.82 \times 10^{-2}$ |
| RAAI Distribution | $2.14 \times 10^{-5}$ | $4.40 \times 10^{-5}$ | $3.97 \times 10^{-2}$ | $4.12 \times 10^{-2}$ |
| Pretraining Skipped | $7.63 \times 10^{-7}$ | $1.86 \times 10^{-5}$ | $3.89 \times 10^{-5}$ | $3.56 \times 10^{-4}$ |
| Pretraining (Unregularized) | $1.64 \times 10^{-7}$ | $3.20 \times 10^{-6}$ | $1.02 \times 10^{-5}$ | $3.73 \times 10^{-5}$ |
| Pretraining (with Thm. 3.1) | $\mathbf{7.86 \times 10^{-8}}$ | $\mathbf{8.86 \times 10^{-7}}$ | $\mathbf{5.27 \times 10^{-7}}$ | $\mathbf{7.87 \times 10^{-6}}$ |

*Table 2.* Minimum and mean (30 samples) MSE error approximating $y = \sin(x)$ and $y = \tanh(3x)$.

| Training Type | Min $\sin(x)$ | Min $\tanh(3x)$ | Mean $\sin(x)$ | Mean $\tanh(3x)$ |
|---|---|---|---|---|
| Default Network (Kaiming) | $4.50 \times 10^{-5}$ | $5.75 \times 10^{-5}$ | $1.15 \times 10^{-1}$ | $1.96 \times 10^{-1}$ |
| RAAI Distribution | $3.59 \times 10^{-5}$ | $1.09 \times 10^{-5}$ | $3.63 \times 10^{-2}$ | $2.31 \times 10^{-2}$ |
| Pretraining Skipped | $1.96 \times 10^{-7}$ | $1.07 \times 10^{-6}$ | $1.93 \times 10^{-5}$ | $8.38 \times 10^{-5}$ |
| Pretraining (Unregularized) | $\mathbf{4.41 \times 10^{-8}}$ | $1.49 \times 10^{-7}$ | $1.47 \times 10^{-5}$ | $3.81 \times 10^{-4}$ |
| Pretraining (with Thm. 3.1) | $5.06 \times 10^{-8}$ | $\mathbf{6.82 \times 10^{-8}}$ | $\mathbf{2.21 \times 10^{-7}}$ | $\mathbf{8.42 \times 10^{-7}}$ |

stantially outperform standard initialization and training procedures. Especially striking is the increased number of linear regions in Figure 7. Like the previous experiments, these networks were trained using Adam for 1000 epochs at a learning rate of 0.001, and the minimum over 30 trainings was taken. The parameterization switch was made halfway through training. These experiments use a slightly improved pretraining construction (see Appendix A.1): the input domain is modified to be $[-1, 1]$, and the network is adjusted so that the $t_1$ and $t_2$ neurons compose "v" shapes rather than triangles, which allows the network to extrapolate better outside $[-1, 1]$. With multiple copies of the network, we implement the calculation of the $d$-th hidden layer as one large combined matrix. Since the hidden neurons of each subnetwork do not connect, the combined matrix takes on a sparse block diagonal structure with a block size of 4. When we switch parameterizations, we release the block diagonal constraint and allow gradient descent to fill in the zeroed weights.

We make a few notes on our preliminary extended technique. First, the class of non-convex functions that can be represented as a difference of convex functions is broad. Zhang et al. (2018) gives an iterative process to decompose an arbitrary ReLU network into a difference of piecewise linear convex functions. Additionally, all functions with bounded second derivatives (or bounded eigenvalues of their Hessian matrix) can be expressed as a difference of two convex functions. Finally, we note that these types of decompositions are not necessarily unique; for example, while $x^3 - x$ can be expressed as $(x^3 - x + 3x^2) - 3x^2$, $3x^2$ is merely the minimal parabola needed to make the second derivative positive on $(-1, 1)$, and larger coefficients could also be used.

Even though our original technique is not naturally designed for multivariate functions and has to be extended by incorporating it as an activation function in a larger network, we note that doing so has some theoretical backing. The multiplication gate and polynomial interpolation network discussed in the related work (Section 2.1) rely on linear combinations of squared terms. Since our networks can produce $x^2$, an arbitrary width and depth arrangement of our networks would retain the ability to perform polynomial interpolation. Secondly, the Kolmogorov-Arnold representation theorem (Kolmogorov, 1957; Arnold, 1957; 1959) gives a guarantee that every continuous multivariate function can be represented by a neural network-like structure that is built using only addition and arbitrary continuous one-dimensional activation functions. Essentially, it states that a technique that only works for univariate functions can still be sufficient for representing multivariate functions when deployed in a larger network. The recently popularized Kolmogorov-Arnold Networks (KAN) (Liu et al., 2024) take their inspiration from this theorem, extending their networks to arbitrary width and depth, but restricting the activations to splines. Even though this does not strictly follow the theorem, they still find empirical success. Similarly, in the case of this work, the functions that the triangular parameterization can directly represent are not dense, as they require a certain degree of self-similarity. Nonetheless, when assembled into a larger network, this may not pose a serious problem.

## 6. Preliminary Image Classification Results

Since our extended technique can be used to replace any dense layers in arbitrary architectures, we can use it in con-

*Table 3.* Minimum errors on unseen points from training on sparse data.

| Training Type | Min $x^3$ | Min $x^{11}$ | Min $\sin(x)$ | Min $\tanh(3x)$ |
|---|---|---|---|---|
| Default Network (Kaiming) | $2.41 \times 10^{-4}$ | $2.14 \times 10^{-3}$ | $2.27 \times 10^{-5}$ | $1.60 \times 10^{-4}$ |
| Pretraining (with Thm. 3.1) | $\mathbf{5.65 \times 10^{-6}}$ | $\mathbf{6.53 \times 10^{-4}}$ | $\mathbf{7.92 \times 10^{-7}}$ | $\mathbf{5.09 \times 10^{-6}}$ |

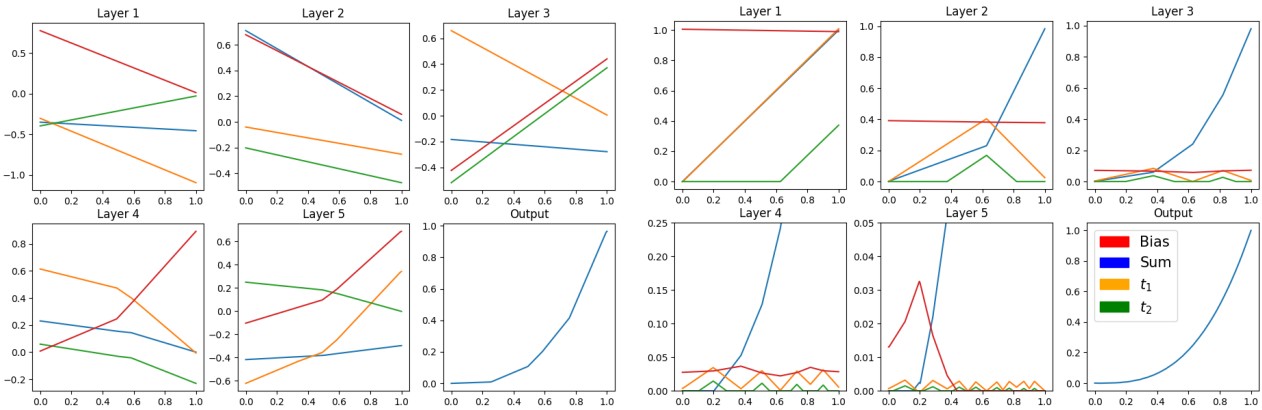

*Figure 5.* Neuron outputs of a default (Kaiming-initialized) network (left) versus a pretrained variant of our network (right). Notice that the first two layers of the default network introduce no linear regions - none of the lines cross zero. Any infinitesimal adjustment to the slopes or biases of the lines would not make such an intersection occur. Therefore, the number of linear regions generated by the network cannot be a local property, and we can expect gradient-based optimization to struggle at maximizing the linear region count. Our method uses this non-locality to our advantage. The pretraining phase finds a low-loss solution where $2^d$ linear regions are generated, which guarantees a neighborhood of parameter space where $2^d$ regions can be maintained while training in the standard parameterization.

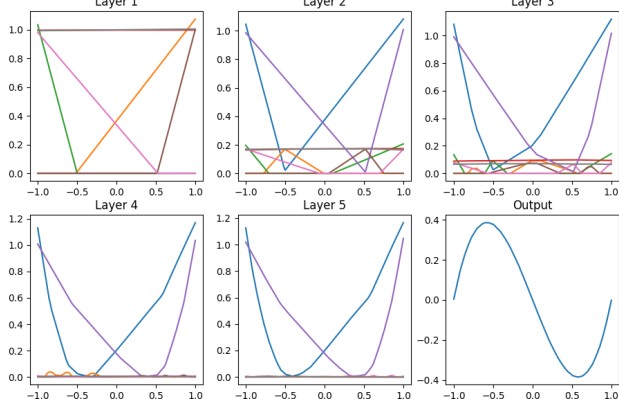

*Figure 6.* Approximation of $y = x^3 - x$ by difference of pretrained components, achieving a loss of $5.52 \times 10^{-7}$. A standard $8 \times 5$ network yielded a larger loss of $8 \times 10^{-6}$.

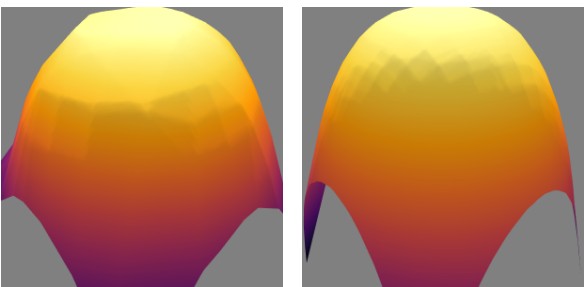

*Figure 7.* Approximations of $z = \sqrt{x^2 + y^2}^3$ using an $8 \times 5$ regular network (left) and a union of two of our pretrained components (right). Losses are $1.5 \times 10^{-4}$ and $3.5 \times 10^{-6}$, respectively, demonstrating a nearly two orders of magnitude improvement using our techniques.

texts beyond regression. In our final experiment, we replace the densely connected classifier of VGG-16 (Simonyan & Zisserman, 2014), and then retrain the network on ImageNet (Deng et al., 2009). We initialize with PyTorch's pretrained convolutional weights both to speed up training and to better isolate the effects of modification to the dense layers. Since VGG-16's dense layers have a width of 4096, we decided to leave the matrices corresponding to our subnetworks in block diagonal format, rather than allowing them to become dense. This way, our technique only adds a negligible amount of extra operations (roughly 0.5%) and does not leave the majority of weights to be filled in by unassisted gradient descent.

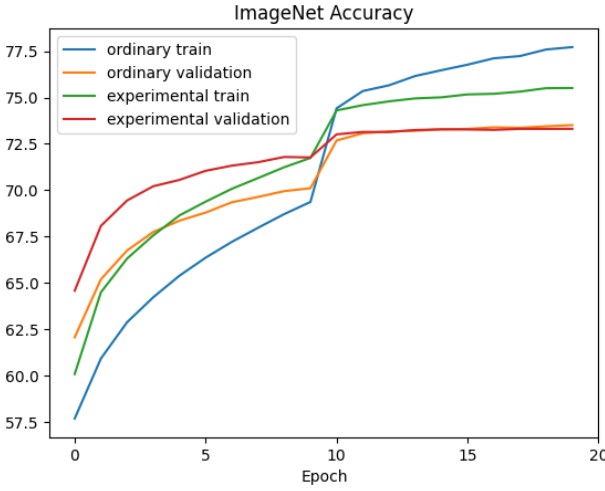

*Figure 8.* Retraining the classifier of a VGG-16 model on ImageNet. The ordinary network uses a learning rate of $1 \times 10^{-4}$, and our experimental network uses $1 \times 10^{-5}$. At epoch 10, both learning rates decay by a factor of 10, and we switch the parameterization in the experimental network. Both networks retrain slightly higher than PyTorch's original reported accuracy of 73.3%.

Figure 8 shows that our network has an advantage early in training, but the fully trained networks achieve comparable accuracy. This is perhaps unsurprising: ordinarily initialized and trained networks are already extremely good at classification problems despite the pathological properties of random initialization. This result suggests that highly precise representations of decision boundaries may not be critical to performance for such classification problems. In support of this, we noticed that it did not seem to matter if we switched parameterizations with our technique, even though doing so was critical to our performance on the 1D regression problems. A practical-scale regression task may be needed for our method to display a significant advantage; we plan such examples as future work.

## 7. Concluding Remarks

This paper focused on exploiting the potential computational complexity advantages neural networks offer for the problem of efficiently learning nonlinear functions; in particular, compelling ReLU networks to approximate functions with exponential accuracy as network depth is linearly increased. Our results showed improvements of one to several orders of magnitude in using our initialization and pretraining strategy to train ReLU networks to learn various nonlinear functions, including non-convex and multi-dimensional functions. This finding is particularly powerful since random initialization and gradient descent are not likely to produce an efficient solution on their own, even if it can be proven to exist in the set of sufficiently sized ReLU networks.

We anticipate the continued development of theoretical improvements to this work. Of particular importance, we think, is either finding a dense set of one-dimensional functions we can represent efficiently with deep networks, or finding a method that more naturally represents multidimensional functions. While works like the KAN show it is possible to perform well without those things, we believe them to be important. The fact that our current technique is limited to creating sparse block diagonal matrices is indicative of a need to determine how to use all the weights of dense layers. Another interesting avenue for increasing the technique's expressiveness could be to consider the generation of fractals. We disallow such behavior by using Theorem 3.1 to enforce differentiability in the hopes of regularization, but there may be more sophisticated approaches to function modeling that can generate fractals in a controlled manner. Given that the technique in its current form is flexible enough to go into any architecture, it is also important to explore possible use cases. We have shown that it will likely not lead to significant improvements on classification tasks, but the technique may shine when given an appropriate utility-scale regression task. Overall, we are hopeful that future works by our group and others will help illuminate a complete theory for harnessing the potential exponential power of depth in ReLU and other classes of neural networks.

## Impact Statement

This paper presents work whose goal is to enable more efficient neural networks. Future advances along this line could enable the use of much smaller networks in practical applications, which could substantially mitigate the rapidly growing issue of energy usage in large learning systems.

## Acknowledgments

M.M. would like to thank Xander Neuwirth for insightful conversations. D.H. was supported in part by the U.S. National Science Foundation under award number 2442853.

This work was authored in part by the National Renewable Energy Laboratory (NREL), operated by Alliance for Sustainable Energy, LLC, for the U.S. Department of Energy (DOE) under Contract No. DE-AC36-08GO28308. This work was supported by the Laboratory Directed Research and Development (LDRD) Program at NREL. The views expressed in the article do not necessarily represent the views of the DOE or the U.S. Government. The U.S. Government retains and the publisher, by accepting the article for publication, acknowledges that the U.S. Government retains a nonexclusive, paid-up, irrevocable, worldwide license to publish or reproduce the published form of this work, or allow others to do so, for U.S. Government purposes.

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

# A. Algorithm and Theory

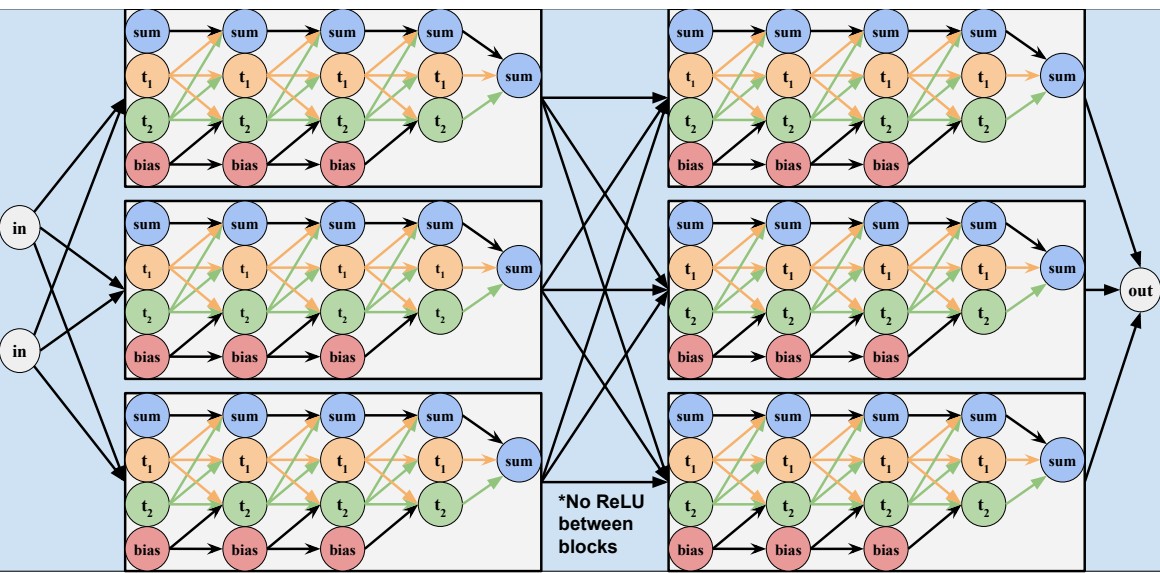

*Figure 9.* Extending our results into a larger network. ReLU should not be used in the layers between the blocks. Colors correspond to Figure 2.

## A.1. Initialization and Pretraining Algorithm

The initialization step of our algorithm is to generate a vector $A = [a_0, a_1, ...a_n]^T$, where each $a_i$ is randomized in $(0, 1)$. Given this $A$, the pretraining step of our algorithm (in the experiments in Section 4) sets the weights of the input ($I$), hidden ($H_i$, $1 \leq i \leq n-1$), and output ($O$) layers of the network as follows:

$$I(x) = \begin{bmatrix} x \\ x \\ x \\ 0 \end{bmatrix} + \begin{bmatrix} 0 \\ 0 \\ -a_0 \\ 1 \end{bmatrix}$$

$$H_i(x) = \begin{bmatrix} 1 & \pm[S_i/a_i & -S_i/(a_i - a_i^2)] & 0 \\ 0 & S_i/a_i & -S_i/(a_i - a_i^2) & 0 \\ 0 & S_i/a_i & -S_i/(a_i - a_i^2) & -S_i a_{i+1} \\ 0 & 0 & 0 & S_i \end{bmatrix} \times ReLU(H_{i-1}(x))$$

$$O(x) = \begin{bmatrix} 1 & S_n/a_n & -S_n/(a_n - a_n^2) & 0 \end{bmatrix} \times ReLU(H_{n-1}(x)),$$

where $S_i$ can either be chosen independently or chosen based on $A$. In the latter case, Equation 3.1 gives $S_i = s_i/s_{i-1} = (1 - a_i)a_{i+1}$. For this version, one must decide whether to add or subtract triangle waves from the sum neuron based on whether the target function is convex or concave, respectively. The version of the algorithm used in Section 5 onward does not have this stipulation and always follows Equation 3.1. Additionally, it expands the input domain to $[-1, 1]$ and flips the triangles upside down to compose "v"-shaped functions instead:

$$I(x) = \begin{bmatrix} 0 \\ \frac{-0.5}{a_0} \\ \frac{0.5}{1-a_0} \\ 0 \end{bmatrix} x + \begin{bmatrix} 1 \\ 1 - \frac{0.5}{a_0} \\ 1 - \frac{0.5}{1-a_0} \\ 1 \end{bmatrix}$$

$$H_i(x) = \begin{bmatrix} 1 & 1 & 1 & -1 \\ 0 & a_{i+1} - 1 & a_{i+1} - 1 & a_i(1 - a_{i+1}) \\ 0 & \frac{a_i(1-a_{i+1})}{1-a_i} & \frac{a_i(1-a_{i+1})}{1-a_i} & \frac{-a_i^2(1-a_{i+1})}{1-a_i} \\ 0 & 0 & 0 & a_i(1 - a_{i+1}) \end{bmatrix} \times ReLU(H_{i-1}(x))$$

$$O(x) = \begin{bmatrix} 1 & 1 & 1 & -1 \end{bmatrix} \times ReLU(H_{n-1}(x)).$$

This assignment of $I$, $H$, and $O$ is used in each iteration of the pretraining algorithm to set the weights of each layer before the forward pass. The backward pass can then propagate the gradients back through this weight setting procedure to update the vector $A$ containing the triangle peaks.

After pretraining, the weights can then be initialized once more based on the learned vector $A$, and then updated directly using regular gradient descent in a second round of optimization. Full pseudocode is listed in Algorithm 1.

---

**Algorithm 1** Initialization and Pretraining

---

$A \leftarrow \text{Random}((0,1)^n)$
**while** Epochs $> 0$ **do**
$\quad$ Network $\leftarrow$ Set_Weights($A$) {Set weights as above each iteration}
$\quad$ Loss $\leftarrow (\text{Network}(x) - y)^2$
$\quad$ Network-Gradient $\leftarrow$ Derivative(Loss, Network) {Regular Backpropagation}
$\quad$ A-Gradient $\leftarrow$ Derivative(Network, $A$) {Backpropagate through weight setting}
$\quad$ Gradient $\leftarrow$ Network-Gradient $\times$ A-Gradient
$\quad$ $A \leftarrow A - \epsilon \times$ Gradient {Update A, Not the network}
**end while**

---

### A.2. Terminology

Before presenting further formal mathematical details of our method, we first briefly review a few pieces of basic ReLU network terminology used in the paper. The reader is referred to Chmielewski-Anders (2020) for an excellent alternative presentation of these terms.

An *activation pattern* is a boolean mask that tracks which neurons in a network have their output zeroed by ReLU activations. The *activation regions* of a ReLU network are connected (and can be shown to be convex) sets of inputs on which the activation pattern is constant. Since the action of the ReLU activation function is constant, the network output over an activation region is equivalent to the case where there is no activation function and the associated zeroed neurons are absent; therefore, the network output behaves linearly over the inputs in the activation region. Relatedly, a *linear region* is a set of inputs on which the network output behaves linearly with respect to its inputs. It may consist of multiple neighboring activation regions. Another important concept is the *boundary* of a neuron, as described in Rolnick & Kording (2020): the set of inputs for which the neuron outputs 0, independently of ReLU. The boundary of a neuron is precisely the boundary of the activation regions it adds to the network. Chen & Ge (2024) refers to this set as the *characteristic activation boundary* since these are the boundaries of the activation regions.

We note that although works like ours are theoretical papers that leverage these concepts, studying these ideas can lead to interesting applied learning research. For instance, Rolnick & Kording (2020) leverages theoretical works like those cited in our related work discussion for an exciting privacy application: it is proven that a ReLU network's output can often be provably used to reverse engineer the architecture of a ReLU network, up to isomorphism.

### A.3. Necessary Conditions for Differentiability

For convenience, we first restate the functions defined in the main body of the paper.

$$T_i(x) = \begin{cases} \frac{x}{a_i} & 0 \leq x \leq a_i \\ 1 - \frac{x-a_i}{1-a_i} & a_i \leq x \leq 1 \end{cases}$$

$$T_i'(x) = \begin{cases} \frac{1}{a_i} & 0 < x < a_i \\ \frac{1}{1-a_i} & a_i < x < 1 \end{cases}$$

$$W_i(x) = \bigcirc_{j=0}^{i} T_j(x) = T_i(T_{i-1}(...T_0(x)))$$

$$F(x) = \sum_{i=0}^{\infty} s_i W_i(x).$$

The goal of this section is to show how to select the $s_i$ based on $a_i$ in a manner where the derivative $F'(x)$ is defined and continuous on all of $[0, 1]$. We begin by assuming that

$$F'(x) = \sum_{i=0}^{\infty} s_i W_i'(x).$$

We will see that the resulting choice of $s_i$ ensures uniform convergence of the derivative terms, so that the derivative of the infinite sum is indeed infinite sum of the derivatives. Fortunately, the left and right derivatives $F_+'(x)$ and $F_-'(x)$ already exist everywhere, since each bend in each $W_i$ (where the full derivative of $F$ is undefined) is assigned the slope of the line segment to its left or right, respectively. The $s_i$ scaling values will have to be chosen appropriately so that $F_+'(x)$ and $F_-'(x)$ are equal for all bend points.

Notationally, we will denote the sorted $x$-locations of the peaks and valleys of $W_i(x)$ by the lists $P_i = \{x : W_i(x) = 1\}$ and $V_i = \{x : W_i(x) = 0\}$. We will use the list $B_i$ to reference the locations of all non-differentiable points, which we refer to as bends. $B_i := P_i \cup V_i$. $f_i(x) = \sum_{n=0}^{i-1} s_n W_n(x)$ will denote finite depth approximations up to but not including layer $i$. The error function $E_i(x) = \sum_{n=i}^{\infty} s_n W_n(x) = F(x) - f_i(x)$ will represent the error between the finite approximation and the infinite depth network. This odd split around layer $i$ makes the proofs cleaner.

Figure 10 highlights some important properties about composing triangle functions. Peaks alternate with valleys. Peak locations in one layer become valleys in the next. Valleys in one layer remain valleys in all future layers since 0 is a fixed point of each $T_i$. To produce $W_i$, each line segment of $W_{i-1}$ becomes a dilated copy of $T_i$. Each triangle function has two distinct slopes, $1/a_i$ and $-1/(1-a_i)$, which are dilated by the chain rule during the composition. On negative slopes of $W_{i-1}$, the input to layer $i$ is reversed, so those copies of $T_i$ are reflected. Due to the reflection, the slopes of $W_i$ on each side of a peak or valley are proportional. Alternatively, one could consider that on each side of a peak in $W_{i-1}$, there is a neighborhood of points that are greater than $a_i$, and are composed with the same line segment of $T_i$ that has slope $-1/(1-a_i)$. Either way, it is important to note that the slopes on each side of a bend scale identically during each composition.

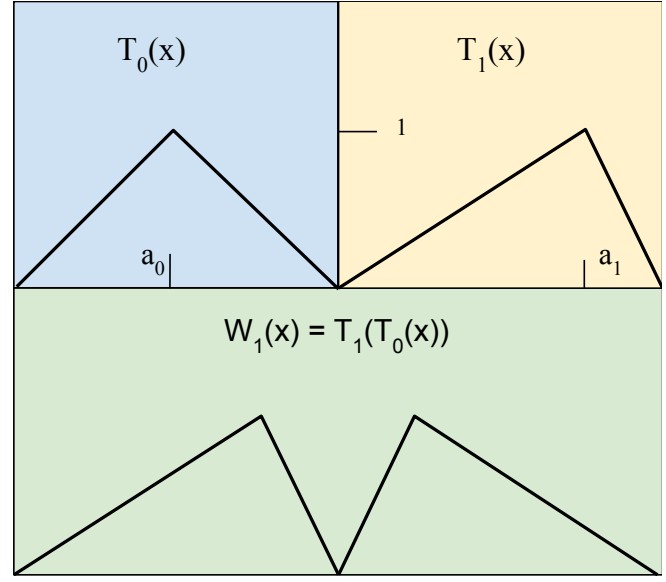

Figure 10. Triangle functions $T_0$ and $T_1$, and the triangle wave resulting from their composition. Note how $T_1$ is reflected in $W_1$.

Before we begin reasoning about $F'(x)$, it can simplify the analysis to only consider the derivative of the error function $E'(x)$.

**Lemma A.1.** *for $x \in P_i$, $F'(x)$ is defined if and only if $E_i'(x)$ is defined.*

*Proof.* All of $W_n(x)$ for $n < i$ are differentiable at $x \in P_i$ since $x$ will lie in the interior of a linear region of $W_n$. Therefore, $f_i'(x) = \sum_{n=0}^{i-1} s_n W_n'(x)$ exists at these points. Since $E_i' + f_i' = F'$, $F'(x)$ is defined if and only if $E_i'(x)$ is defined. $\square$

Thanks to the previous lemma, we only need to work with $E_i$. Here we compute the right derivative $(E_i)_+'(x)$ of the error function at a point $x$. The left derivative will only be different by a constant factor.

**Lemma A.2.** *For all $x \in P_i$, $E_+'(x)$ and $E_-'(x)$ are proportional to*

$$s_i - \frac{1}{1 - a_{i+1}} \left( s_{i+1} + \sum_{n=i+2}^{\infty} s_n \prod_{k=i+2}^{n} \frac{1}{a_k} \right). \tag{4}$$

*Proof.* Let $x_k$ be some point in $P_i$, and let $k$ be its index in any list it appears in. To calculate the value of $E_+'(x_k) = \sum_{n=i}^{\infty} s_n (W_n)_+'(x_k)$, we will have to find the slope of the linear intervals to the immediate right of $x_k$ for all $W_i$. We will use $R_x$ to represent $W_{i+}'(x_k)$. The first term in the sum will be $R_x s_i$. Since the derivatives of composed functions will multiply from the chain rule, so the value of the next term is $W_{i+1}'(x_k) = T_{i+1}'(W_i(x_k)) R_x$. $T_{i+1}$ has two linear segments, giving two slope possibilities to multiply by. The correct one to choose is $-1/(1 - a_{i+1})$ because it is "active" around $x_k$ ($x_k$ is a peak of $W_i$, so $W_i(x_k) > a_{i+1}$ for $x \in (B_{i+1}[k-1], B_{i+1}[k+1])$). This gives $W_{i+1}'(x_k) = -R_x \frac{s_{i+1}}{(1-a_{i+1})}$. Note that the second term has the opposite sign as the first.

For all remaining terms, since $x_k$ was in $P_i$, it is in $V_j$ for $j > i$. For $x \in (B_{j+1}[k-1], B_{j+1}[k+1])$, $W_j(x) < a_{j+1}$ and the chain rule applies the slope $1/a_{j+1}$. Since this slope is positive, every remaining term continues to have the opposite sign as the first term. Summing up all the terms with the coefficients $s_i$, and factoring out $R_x$ will yield the desired formula. Note that this same derivation applies to the left sided derivatives as well because the "active" slopes of $T_{i+1}(W_i)$ are all the same whether a bend in $W_i$ is approached from the left or the right. The initial slope constant $L_x$ will just be different. $\square$

**Lemma A.3.** *If $E_+'(x) = E_-'(x)$, $E'(x)$ must be equal to 0.*

*Proof.* Let $S$ represent Equation 4, and $R$ and $L$ be the constants of proportionality for the directional derivatives. If $E_+' = E_-'$, then $R_x S = L_x S$ for all $x \in P_i$. Since $W_i$ is comprised of alternating positive and negatively sloped line segments, $R_x$ and $L_x$ have opposite signs. The only way to satisfy the equation then is if $S = 0$. Consequently, $E'(x) = 0$ for all $x \in P_i$. $\square$

The following lemma shows that to calculate the derivative at of $F(x)$ for any bend point $x$, one needs only to compute the derivative of the finite approximation $f_i$ (which excludes $W_i$). This will be useful later for proving other results.

**Lemma A.4.** *For all $x \in P_i$:*

$$F'(x) = f_i'(x) = \sum_{j=0}^{i-1} s_j W_j'(x). \tag{5}$$

*Proof.* From the previous lemma we have $E'(x) = 0$ whenever the directional derivatives are equal. $F(x) = \sum_{j=0}^{i-1} s_j W_j(x) + E(x)$. The first $i - 1$ terms are differentiable at the points $P_i$ since those points lie between the discontinuities in $B_{i-1}$. Therefore $F'(x)$ is defined and can be calculated using the finite sum. A visualization of this lemma is provided in Figure 11. $\square$

We now prove our main theorem, which shows that there is a way to sum the triangular waveforms $W_i$ so that the resulting approximation converges to a continuously differentiable function. The idea of the proof is that much of the formula for $E'(x)$ will be shared between two successive generations of peaks. Once they are both valleys, they will be treated the same by the remaining compositions, so the sizes of their remaining discontinuities will need to be proportional.

**Theorem 3.1.** *$F'(x)$ is continuous on $[0, 1]$ only if the scaling coefficients are selected based on $a_i$ according to:*

$$s_{i+1} = s_i(1 - a_{i+1})a_{i+2}.$$

*Proof.* Rewriting Equation 4 (which is equal to 0) for layers $i$ and $i+1$ in the following way:

$$s_i(1 - a_{i+1}) = s_{i+1} + \frac{1}{a_{i+2}} \left( s_{i+2} + \sum_{n=i+3}^{\infty} s_n \prod_{k=i+3}^{n} \frac{1}{a_k} \right)$$

$$s_{i+1}(1 - a_{i+2}) = s_{i+2} + \sum_{n=i+3}^{\infty} s_n \prod_{k=i+3}^{n} \frac{1}{a_k}$$

allows for a substitution to eliminate the infinite sum

$$s_i(1 - a_{i+1}) = s_{i+1} + \frac{1 - a_{i+2}}{a_{i+2}} s_{i+1}.$$

Collecting all the terms gives

$$s_{i+1} = \frac{s_i(1 - a_{i+1})}{1 + \frac{1 - a_{i+2}}{a_{i+2}}},$$

which simplifies to the desired result. $\square$

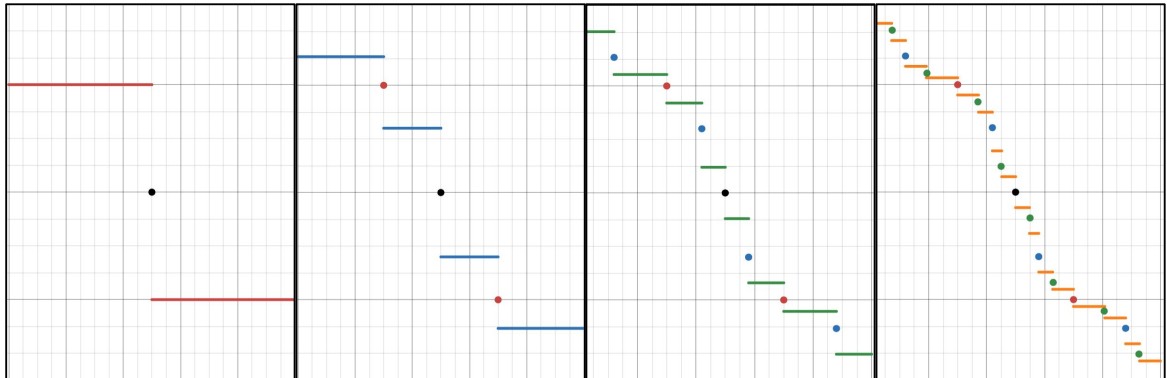

*Figure 11.* The derivatives of the first few stages of approximation. Notice that each time a constant segment "splits," the two neighboring segments adjacent to the split monotonically converge back to the original value (marked with a colored point corresponding to the step it originates from).

### A.4. Sufficiency for differentiability

This form of the scaling equation derived in the previous section is rather interesting. Since the ratio of two successive scaling terms is $(1 - a_{i+1})a_{i+2}$, factors of both $a_i$ and $1 - a_i$ are present in $s_i$. This has a few implications. Firstly, if any $a_i$ is 0 or 1, subsequent scales will all be 0, essentially freezing the corresponding neural network at a finite depth. Secondly, having both $a_i$ and $1 - a_i$ will cancel whichever slope multiplier $T_i$ contributes to $W_i$ at each point $x$, leaving behind the other term, which is less than 1.

If we ensure each $a_i$ is bounded away from 0 or 1 by being drawn from the an interval such that $c < a_i < 1 - c$ for some $0 < c < 0.5$, then the maximum value of $W_i'$ is always upper bounded by $(1 - c)^i$, which is sufficient for uniform convergence. This condition also means that the bends points (and thus the activation regions of the corresponding neural network) will become dense in $[0, 1]$, as each region is partitioned (at worst) in a $c : 1 - c$ ratio.

Lastly, we can show that bounds on $a_i$ and our choice of scaling values are sufficient for the existence of $F'$ on the bend points, in addition to being necessary for the existence of the derivative on the bend points, our choice of scaling is sufficient when $a_i$ are bounded away from 0 or 1.

**Theorem A.6.** *On bend points $x$, $F'(x)$ exists if we can find $c > 0$ such that $c \le a_i \le 1 - c$ for all $i$ and choose all $s_i$ according to Equation 3.1.*

*Proof.* We begin by considering Equation 4 for layer $i$ (it equals 0 by Theorem A.3).:

$$s_i = \frac{1}{1 - a_{i+1}} \left( s_{i+1} + \sum_{n=i+2}^{\infty} s_n \prod_{k=i+2}^{n} \frac{1}{a_k} \right).$$

We will prove our result by substituting Equation 3.1 into this formula, and then verifying that the resulting equation is valid. First we would like to rewrite each occurrence of $s$ in terms of $s_i$. Equation 3.1 gives a recurrence relation. Converting it to an non-recursive representation, we have:

$$s_n = s_i \left( \prod_{j=i+1}^{n} 1 - a_j \right) \left( \prod_{k=i+2}^{n+1} a_k \right). \tag{6}$$

When we substitute this into Equation 4, three things happen: each term is divisible by $s_i$ so $s_i$ cancels out, every factor in the product except the last cancels, and $1 - a_{i+1}$ cancels. This leaves

$$1 = a_{i+2} + (1 - a_{i+2})a_{i+3} + (1 - a_{i+2})(1 - a_{i+3})a_{i+4} + \dots = \sum_{n=i+2}^{\infty} a_n \prod_{m=i+2}^{n-1} (1 - a_m). \tag{7}$$

This equation has a meaningful interpretation that is important to the argument. 1 is the full size of the initial derivative discontinuity at a point in $P_i$, and each term on the other side represents proportionally how much the discontinuity is closed for each triangle wave that is added. Every time a wave is added, it subtracts the first term appearing on the right hand side. The following argument shows that each term of the sum on the right accounts for a fraction (equal to $a_i$) of the remaining discontinuity, guaranteeing its disappearance in the limit. Inductively, we can show:

$$1 - \sum_{n=i+2}^{j} a_n \prod_{m=i+2}^{n-1} (1 - a_m) = \prod_{m=i+2}^{j} (1 - a_m). \tag{8}$$

In words, this means that as the first term appearing on the right in Equation 7 is repeatedly subtracted, that term is always equal to $a_n$ times the left side. As a base case, we have $(1 - a_{i+2}) = (1 - a_{i+2})$. Assuming the above equation holds for all previous values of $j$,

$$1 - \sum_{n=i+2}^{j+1} a_n \prod_{m=i+2}^{n-1} (1 - a_m)) = 1 - \sum_{n=i+2}^{j} a_n \prod_{m=i+2}^{n-1} (1 - a_m)) - a_{j+1} \prod_{m=i+2}^{j} (1 - a_m)),$$

using the inductive hypothesis to make the substitution

$$\prod_{m=i+2}^{j} (1 - a_m)) - a_{j+1} \prod_{m=i+2}^{j} (1 - a_m)) = \prod_{m=i+2}^{j+1} (1 - a_m)).$$

Since all $c < a_i < 1 - c$, the size of the discontinuity at the points $P_i$ is upper bounded by the exponentially decaying series $(1 - c)^n$, which approaches zero. $\square$

### A.5. Error Decay

**Lemma A.7.** *The ratio $s_{i+2}/s_i$ is at most $0.25$.*

*Proof.* by applying Equation 3.1 twice, we have

$$s_{i+2} = s_i(1 - a_{i+1})(1 - a_i + 2)a_{i+2}a_{i+3}.$$

To maximize $s_{i+2}$, we choose $a_{i+1} = 0$ and $a_{i+3} = 1$. The quantity $a_{i+2} - a_{i+2}^2$ is a parabola with a maximum of $0.25$ at $a_{i+2} = 0.5$. $\square$

Since each $W_i$ takes values between 0 and 1, its contribution to $F$ is bounded by $s_i$. Since the $s_i$ decay exponentially, one could construct a geometric series to bound the error of the approximation and arrive at an exponential rate of decay.

## A.6. Second Derivatives

Here we show that any function represented by one of these networks that is not $y = x^2$ does not have a continuous second derivative, as it will not be defined at the bend locations. To show this we will sample a discrete series of $\Delta y / \Delta x$ values from $F'(x)$ and show that the limits of these series on the right and left are not the same (unless all $a_i = 0.5$), which implies that $F''(x)$ does not exist (see Figure 12 below). First we will produce the series of $\Delta x$. Let $x$ be the location of a peak of $W_i$, and let $l_n$ and $r_n$ be its immediate neighbors in $B_{i+n}$.

**Lemma A.8.** *If $c < a_i < 1 - c$ for all $i$, we have $\lim_{n\to\infty} r_n = \lim_{n\to\infty} l_n = x$. Furthermore, $r_n, l_n \neq x$ for any finite $i$.*

*Proof.* Let $R$ and $L$ denote the magnitude of $W_i'$ on the left and right of $x$. $x$ is a peak location of $W_i$, so the right side slope is negative and the left is positive. Solving for the location of $T_{i+1}(W_i(x)) = 1$ on each side gives $l_1 = x - (1 - a_{i+1})/L$ and $r_1 = x + (1 - a_{i+1})/R$.

On each subsequent iteration $i + n$ $(n \geq 2)$, $x$ is a valley point and the $\Delta x$ intervals get multiplied by $a_{i+n}$. Since $x$ is a valley point, the right slope is positive and the left is negative. The slope magnitudes are given by $\frac{1}{x - l_n}$ and $\frac{1}{r_n - x}$ since $W_{i+n}$ ranges from 0 to 1 over these spans. Solving for the new peaks again gives $l_{n+1} = x - a_{i+1}(x - l_n)$ and $r_{n+1} = x + a_{i+1}(r_n - x)$. The resulting non-recursive formulas are:

$$x - l_n = \frac{1 - a_{i+1}}{L} \prod_{m=2}^{n} a_{i+m} \text{ and } r_n - x = \frac{1 - a_{i+1}}{R} \prod_{m=2}^{n} a_{i+m}. \tag{9}$$

The right hand sides will never be equal to zero with a finite number of terms since $a$ parameters are bounded away from 0 and 1 by $c$. $\qquad\square$

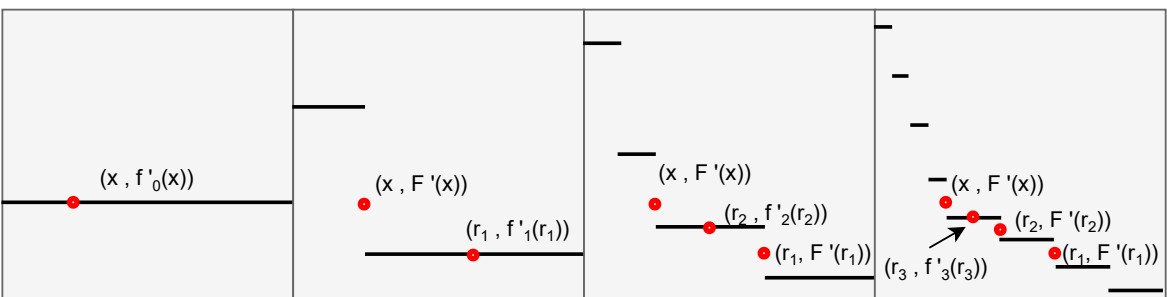

*Figure 12.* An illustration of attempting to calculate the second derivative. The points in the series approaching $x$ from the right are marked. We rely on the fact that at bend points, the first derivative converges back to the value it had at a finite point in the approximation. $a_0 \neq 0.5$ and all other parameters are set to 0.5, which will cause the left and right sets of points to lie on lines with different slopes.

Next we derive the values of $\Delta y$ to complete the proof.

**Theorem A.9.** *$F(x)$ cannot be twice differentiable unless $F(x) = x^2$.*

*Proof.* The points $l_n$ and $r_n$ are all peak locations, Equation 5 gives their derivative values as $f'_{i+n}(r_n)$. In our argument for sufficiency, we reasoned about the sizes of the discontinuities in $f'$ at $x$. Since $l_n$ and $r_n$ always lie on the linear intervals surrounding $x$ as $n \to \infty$, we can get the value of $f'_i(x) - f'_{i+n}(r_n)$ using Equation 8 with the initial discontinuity size set to $Rs_i$ rather than 1. Focusing on the right hand side, we get:

$$f'_i(x) - f'_{i+n}(r_n) = R * s_i \prod_{m=2}^{n} (1 - a_{i+m}).$$

Taking $\Delta y / \Delta x$ gives a series:

$$\frac{R^2 s_i}{(1 - a_{i+1})} \prod_{m=i+2}^{n} \frac{1 - a_m}{a_m}.$$

The issue that arises is that the derivation on the left is identical, except for a replacement of $R^2$ by $L^2$. The only way for these formulas to agree, then, is for $R^2 = L^2$, which implies $a_i = 1 - a_i = 0.5$. Since this argument applies at any layer, then all $a$ parameters must be 0.5 (which approximates $y = x^2$). $\square$

### A.7. Monotonicity and Continuity of Derivatives

Each of the $f_i'$ are composed of constant value segments. We will show that those values are monotonically decreasing (this can be seen in Figure 11). This can extend into the limit to show that $F'$ is monotone decreasing and that $F$ is concave.

**Lemma A.10.** *The function $F(x)$ is concave when all $s_i$ are chosen according to Equation 3.1.*

*Proof.* To establish this, we will introduce the list $Y_i' = [F'(V_i[0]), f_i'(V_i[n]), F'(V_{i+1}[2^i])]$ for $0 \le n \le 2^i$, which tracks the values of $F'$ at the $i^{\text{th}}$ set of valley points. All but the first and last points will have been peaks at some point in their history, so Equation 5 gives the value of those derivatives as $f_i'$.

We establish two inductive invariants. One is that the y-values in the list $Y_i$ remain sorted in descending order. The other is that $Y_i'[n] \ge f_i'(x) \ge Y_i'[n+1]$ for $V_i[n] < x < V_i[n+1]$, indicating that the constant value segments of $f_i$ lie in between the limits in the list $Y_i$. Together, these two facts imply that each iteration of the approximation $f_i$ is concave. This can then be used to prove that their limit $F$ is also concave.

As the base case, $f_0$ is a line with derivative 0, and $V_0$ contains its two endpoints. $Y_0'$ is positive for the left endpoint (negative for right) since on the far edges $F'$ is a sum of a series of positive (or negative) slopes. Therefore, both the points in $Y'$ are in descending sorted order. The second part of the invariant is true since 0 is in between those values.

Consider an arbitrary interval $(V_i[n], V_i[n+1])$ of $f_i$. This entire interval is between two valley points, so $f_i'$ (which has not added $W_i$ yet) is some constant value, which we know from the second inductive hypothesis is in between $Y_i'[n]$ and $Y_i'[n+1]$. The point $x \in P_i \cap (V_i[n], V_i[n+1])$ will have $F'(x) = f_i'(x)$, and it will become a member of $V_{i+1}$. This means we will have $Y_{i+1}[2n] > Y_{i+1}[2n+1] > Y_{i+1}[2n+2]$, maintaining sorted order of $Y'$.

Adding $s_i W_i$ takes $f_i$ to $f_i + 1$ splitting each constant valued interval in two about the points $P_i$, increasing the left side, and decreasing the right side. Recalling from the derivation of Equation 4 all terms but the first in the sum have the same sign, so the limiting values in $Y_i'$ are approached monotonically. Using the first inductive hypothesis, we have on the left interval $Y_i'[n] = Y_{i+1}'[2n] > f_{i+1}' > f_i' = Y_{i+1}'[2n+1]$ and on the right we have $f_i' = Y_{i+1}'[2n+1] > f_{i+1} > Y_i'[n+1] = Y_i'[2n+2]$. And so each constant interval $f_{i+1}$ remains bounded by the limits in $Y_{i+1}'$.

We will now show by contradiction that the limit $F$ of the sequence of concave $f_i$ is also concave. Assume that $F$ is non-concave. Then there exist points $a$, $b$, and $c$ such that $F(b)$ lies strictly below the line connecting the points $(a, F(a))$ and $(c, F(c))$. Let us imagine it is below the line by an amount $\epsilon$. Since at each point $f_i$ converges to $F$, we can find $i_a$ such that $f_{ia}(a) - F(a) < \epsilon/2$, etc..., we take $i = \max(i_a, i_b, i_c)$. Since $f_i(a)$ and $f_i(c)$ are no more than $\epsilon/2$ lower than their limiting values, the entire line connecting $(a, f_i(a))$ and $(c, f_i(c))$ is no more than $\epsilon/2$ lower than the line between $(a, F(a))$ and $(c, F(c))$. $f_i(b)$ is also no more than $\epsilon/2$ higher than $F(b)$, thus $f_i(b)$ must still lie below the line between $(a, f_i(a))$ and $(c, f_i(c))$, making $f_i$ non-concave and producing a contradiction. $\square$

Lastly, we briefly sketch out why $F'$ is continuous. It relies on some of our earlier reasoning. Monotonicity of the derivative makes continuity easy to show, because when $x_1$ within $\delta$ of $x_2$ has $f(x_1)$ within $\epsilon$ of $f(x_2)$, so do all intermediate values of $x$. We can establish continuity of $F'$ at bend points $x$ easily by using Equation 8. We can pick an $i$ so that the constant value segments of $f_i'$ are within $\epsilon$ of $F'(x)$ and then use the next iteration of bend points (since the constant intervals split, but the new segments near $x_n$ converge monotonically towards it) to find $\delta$. In the case of continuity for non bend points $x$, they sit inside a constant-valued interval of $f_i$ for each $i$. We can choose $i$ such that $\sum_{n=i}^{\infty}(1-c)^n < \epsilon/2$ because this series constrains how far derivative values can move in the limit, and then use the constant interval $x$ is situated in to find $\delta$.

### A.8. Activation Region Counting and Zaslavsky's Theorem

In the coming sections, we present the results of using our 4-neuron-wide construction as an activation function on more complicated datasets. It will be useful to know the impact of this approach on the activation region count when assessing how to best spend a network's parameter budget. To build up to it, we will provide an informal introduction of Zaslavsky's theorem (Zaslavsky, 1975). The activation boundaries of ReLU neurons in a single hidden layer network form a hyperplane arrangement. Counting the number of *cells*, which are convex connected components of space separated by the hyperplanes,

is equivalent to determining the number of activation regions produced by the shallow ReLU network. A $D$-dimensional hyperplane arrangement is said to be in *general configuration* when every set of $D$ hyperplanes intersect in a unique point, every set of $D-1$ hyperplanes intersect in a unique line, etc. General configuration is essentially what you would expect on average from setting the orientations and offsets of the hyperplanes randomly (and would thus be applicable to a single hidden layer ReLU network). Zaslavsky's theorem states that

**Theorem A.11.** *When in general configuration, the number of cells $(R\binom{n}{D})$ created by a $D$-dimensional arrangement of $n$ hyperplanes is*

$$R\binom{n}{D} = \sum_{d=0}^{D} \binom{n}{d}.$$

To compute this sum, one would go down to the row of Pascal's triangle corresponding to the number of planes $n$, and sum the first $D$ entries (see Figure 13 for a visual example). It is a known fact that the sequence of the $d$th entry for each row grows to the $d^{\text{th}}$ power. So as the number of hyperplanes tends towards infinity, this sum is dominated by the $D^{\text{th}}$ term, which grows to the power $D$. In other words, the number of linear regions formed by a single hidden layer ReLU network grows exponentially with respect to its input dimension. This property partly explains why neural networks are successful at learning high-dimensional functions, even when other classical methods become intractable.

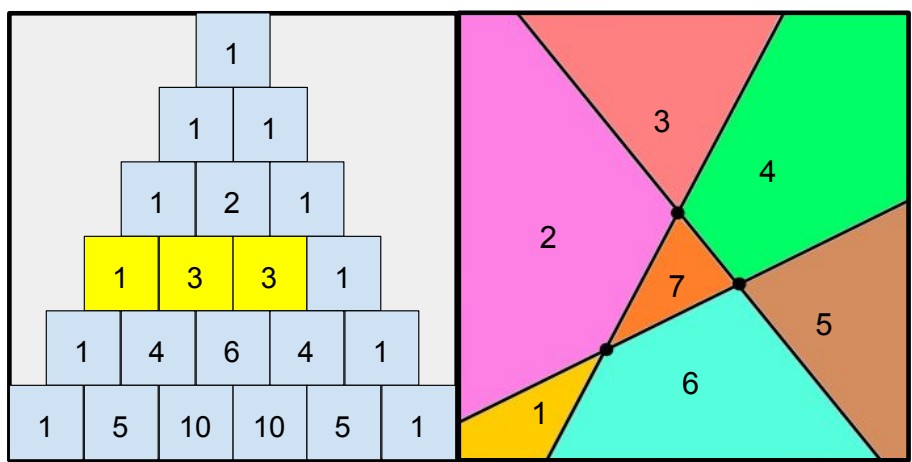

*Figure 13.* 3 lines dividing 2-dimensional space creates 7 cells. This is calculated by going down to row 3 (for the number of lines) and summing the numbers at indices 0, 1, and 2 (for the dimension). Also note that there is one 2D plane with 3 lines and 3 points of intersection in this configuration, corresponding (in that order) to the highlighted entries in Pascal's triangle.

Thanks to the general configuration of the hyperplanes, the number of lower dimensional objects (i.e., points, lines, planes, etc.) are given by the entries in Pascal's triangle. So a different way to phrase Zaslavsky's theorem is that the number of cells created by the arrangement is equal to the sum of all these lower-dimensional objects, or that each lower-dimensional object could be "assigned" a cell. There is a striking similarity in the intuitions behind the recurrence that generates Pascal's triangle, and the calculation of the numbers of lower-dimensional objects created from a hyperplane arrangement. In the case of Pascal's triangle, if one wished to know how to choose 3 items from 6, one could reason that if item 6 is included, 2 items would be chosen from the remaining 5, and if item 6 is excluded, 3 would be chosen from the remaining 5, and therefore $\binom{6}{3} = \binom{5}{2} + \binom{5}{3}$. Now consider the case of 3 planes partitioning 3-space, and the addition of a 4th plane. The total number of lower-dimensional objects would be found by including those that are part of the original 3 planes (1 3-space, 3 planes, 3 lines, 1 point), and adding those that the 4th plane creates (1 plane, 3 lines, 3 points). One could also picture that the new 4th plane "cuts" the existing 3-cells, and to find the number of new cells added, one could examine the projection of the 3 existing planes onto the 4th (shown in Figure 13), and count the 2-cells that appear. Either reasoning leads to $R\binom{n}{D} = R\binom{n-1}{D} + R\binom{n-1}{D-1}$. This similarity between the recurrences means that if the initial conditions can be lined up suitably, Zaslavsky's theorem can be proven by induction over Pascal's triangle.

In the case of our architecture, using our 1-dimensional construction as an activation function will generate $n$ generally configured sets of $m$ parallel hyperplanes. To account for this, a slight modification is made to the recurrence. Adding an additional set of $m$ hyperplanes would add $m$ times as many regions, as each of the lower dimensional objects in the

existing configuration would project onto each of the $m$ additional hyperplanes. The effect that this has on the formula is that $R\binom{n,m}{D} = R\binom{n-1,m}{D} + m \times R\binom{n-1,m}{D-1}$. Inducting over Pascal's triangle (with a base case of $R\binom{n,m}{1} = 1 + nm$) would give the non-recursive formula of $R\binom{n,m}{D} = \sum_{d=0}^{D} m^d \binom{n}{d}$. This means that our architecture (with a single hidden layer of depth 5 convex function blocks) would be generating $32^D$ times as many regions as an ordinary single layer of ReLU, which is a fantastic tradeoff once the layers are wide enough (both so that the largest exponent dominates the sum in Zaslavsky's theorem, and so that only a few neurons would be sacrificed to pay for the new parameters in the activations). Of course, the effect on accuracy will all depend on how well these regions can be utilized.

## B. Additional Experiments

### B.1. Learning Rates

All results in the main body of the paper used a constant learning rate of $10^{-3}$. In this appendix, we considered an ablation study on the learning rate for the task of learning $y = x^3$. As seen in Figure 14, the learning rate we selected was approximately optimal for both our method as well as default network training. We note that for this ablation study, a constant 1000 epochs were run, which explains why both methods perform worse as the learning rate becomes minuscule. At small learning rates, what is really measured is how close of a guess the initialization is to the target function. Our networks are preforming better here simply because they are always outputting convex functions. But this only accounts for losses on the order of $10^{-4}$, which indicates that in the more reasonable learning rate ranges, our pretraining is performing a meaningful function and enabling order-of-magnitude improvements over default network training.

### B.2. Real-World Data and Classification Problems

Here we present a few preliminary results on extending our pretraining technique to classification problems and real-world datasets. The classification problem we chose is the classic two spirals dataset, and we selected the UCI dataset "Concrete Compressive Strength" (Yeh, 1998) for our real-world regression task. The concrete dataset has 8 numerical features that can be used to predict the compressive strength of a concrete sample.

The networks are set up as described before, with our 4-neuron-wide networks acting as 1D to 1D activation functions inside randomly initialized standard linear layers. In these experiments, we use one "hidden layer" of

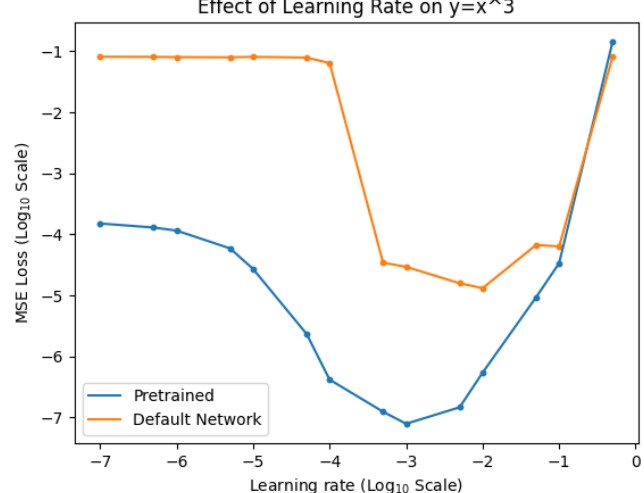

Figure 14. Learning $y = x^3$ at various learning rates for 1000 epochs. The figure shows the log losses of the best model of 30 for networks trained according to the methods in this paper versus random initialization and gradient descent ("Default Network"). For high learning rates, neither learns because the steps are too large. For low learning rates (below $10^{-5}$), the steps are too small, and our networks are likely deriving an implicit advantage from being forcibly initialized to a convex function. Both methods are able to converge for learning rates in between $10^{-4}$ and $10^{-2}$; one could run for more epochs to see a similar advantage of our method for smaller learning rates.

our networks (which we choose to have depth 5 in our tests). The geometric interpretation of this is that we are setting up one-dimensional convex functions oriented in random directions on the input space, and then taking a linear combination of them as the network output. In the case of classification, the loss function is simply swapped for cross entropy.

After our pretraining phase, there are two choices of how to conduct the second phase of training (training matrix entries directly). All the parameters could be freed from constraints ("dense"), or the smaller 4-neuron subnetworks could be kept isolated from each other (but otherwise have their parameters freed). In the case where the 4-neuron networks remain in isolation, the weight matrices of the hidden layers will have a block diagonal structure (block size 4—the width of each subnetwork). Thus, we consider two fair (same number of free parameters) comparisons in this subsection: (1) dense versions of our and Kaiming-initialized networks, and (2) block diagonal versions of our and Kaiming-initialized networks.

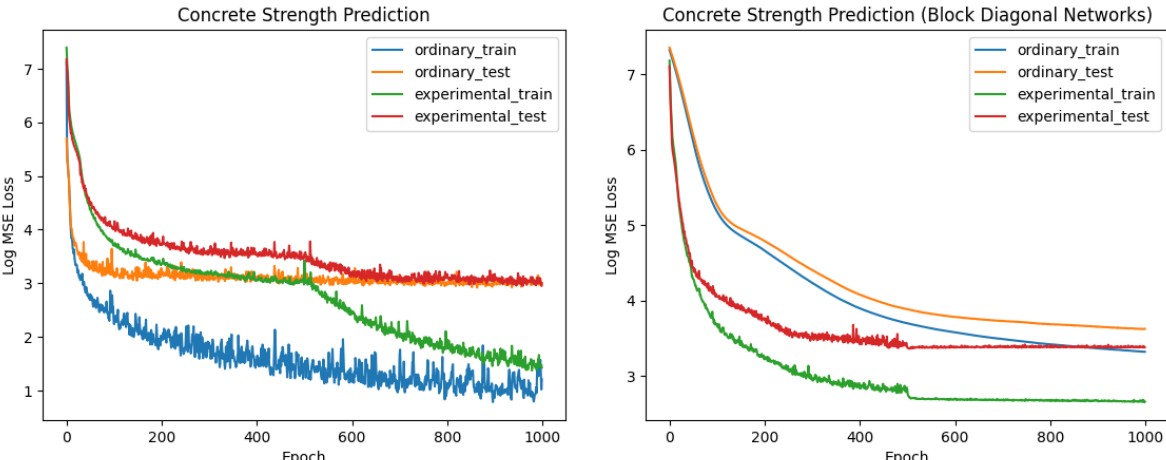

*Figure 15.* Train/test loss plots for Kaiming-initialized ("ordinary") fully-connected networks and our method ("experimental"). Our method has two steps: pretriaining on triangle peaks, followed by the optimization of the raw matrix weights. The switch-off is at epoch 500, hence the associated visible change in the loss curves. The block diagonal variants of the networks (right) are generally worse at the task (test losses 30.1 (ours) and 40.8 (Kaiming)) than the dense variants (losses of 22.3 (ours) and 21.2 (Kaiming)). Our experimental networks outperform by up to an order of magnitude in the block diagonal case. We used 32 of our convex blocks (i.e., all weight matrices are size 128, including for standard comparison networks).

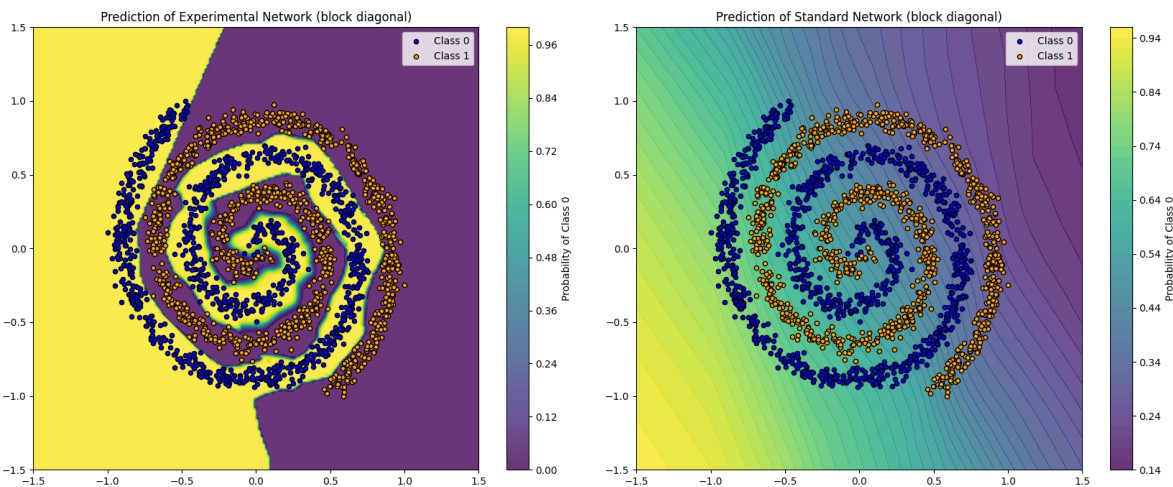

*Figure 16.* Class predictions of block diagonal networks on a standard two-spiral dataset; ours (left) and Kaiming-initialized (right). While the cross-entropy losses are comparable for the dense-matrix variants of both networks, when a block diagonal structure is imposed, Kaiming initialization fails to learn the spiral. Note that the colors of the network class predictions are inverted for visibility. Cross-entropy losses are 0.0032 and 0.63 respectively. We used 16 of our convex blocks, so all weight matrices are 64 by 64.

As discussed below and in the figure captions, we find that while the dense variant of our method ties or is slightly worse than a regular fully-connected network of the same dimensions, in the block diagonal case, our experimental networks significantly outperform their Kaiming-initialized counterparts. This makes sense in light of the experiments from the main body, where we can effectively shape the output of 4-neuron-wide networks better than random initializations, but where ensembles of our networks (see approximation of $x^3 - x$ in Figure 6) get filled in with noise by gradient descent during the second training stage. This again highlights the need for more mathematical developments to provide a better extension into higher-dimensional nonconvex functions.

Nonetheless, while we believe there is much room for future work to improve higher-dimensional results, Figures 15

and 16 already show impressive results using our current approach. In the case of the concrete problem (Figure 15), our experimental network outperforms a standard Kaiming-initialized network when block diagonality is enforced. In the case of the two-spiral classification task (Figure 16), our network is able to learn an accurate decision boundary (left subfigure), whereas a standard network constrained to be block diagonal fails to learn (right subfigure). When dense weight matrices are trained, our networks are almost able to tie with the standard initializations, although convergence can take longer. These results suggest that our method might be quite powerful when its parameter budget is spent on greater width and depth of convex function blocks (as in Figure 9) instead of filling in dense matrices.

Our two-step training approach adds some practical challenges, such as deciding when to switch parameterizations for optimal convergence time. Additionally, since the optimization variables are different, the optimizer will have to restart any momentum or adaptive learning rates when the switch is made, which can sometimes cause loss to temporarily spike. We lowered the learning rate on the second step to avoid this. Further interesting optimizations of our approach (e.g., training another network to inform our algorithm when to switch parameterizations) are imagined as future work.

### B.3. Image Classification on CIFAR-10

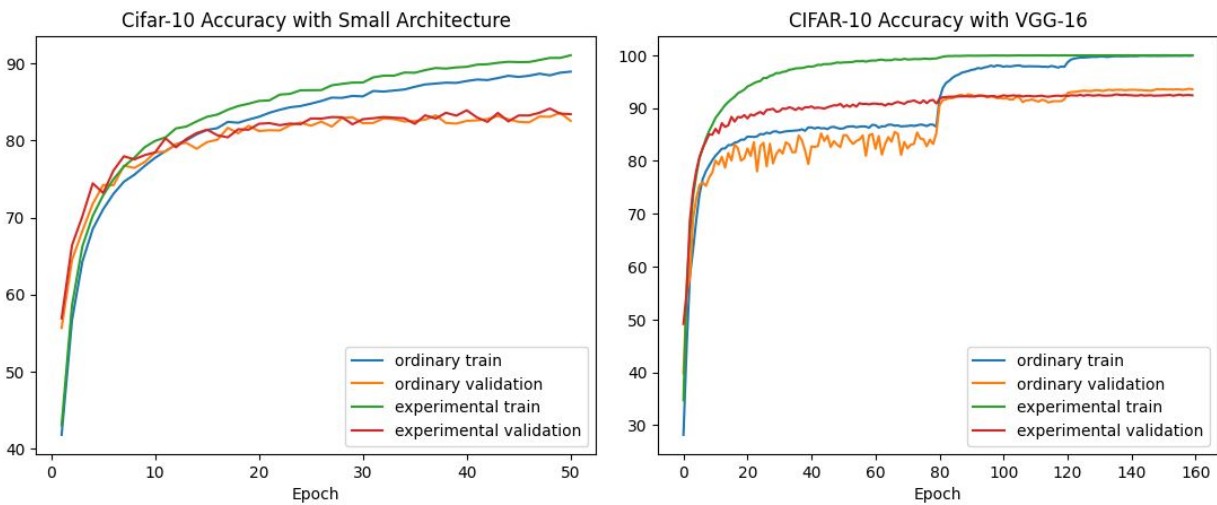

*Figure 17.* Results for CIFAR-10 experiments. Our ("experimental") training and validations are generally comparable to their Kaiming-initialized ("ordinary") counterparts. The small architectures are trained with Adam at a learning rate of 0.001, while the VGG experiments are trained with SGD, following a learning rate schedule with a 10x drop at epochs 80 and 120. The ordinary network uses an initial learning rate of 0.1, and the experimental network uses 0.01. No parameterization switches occur with our method.

A nice feature of our method is that it can be used anywhere dense layers appear in networks, which enables its use in CNN architectures. Here we demonstrate its application for image classification on CIFAR-10 (Krizhevsky et al., 2009). Our networks are inserted into the dense layers of VGG-16, as well as a small-scale CNN architecture of 3 convolutions (channel depths of $3 \rightarrow 32 \rightarrow 64 \rightarrow 128$ with $3 \times 3$ kernels) followed by a single hidden layer dense classifier of 256 neurons, whose width was reduced slightly to 247 to pay for the extra parameters when our method was used. The results we get are comparable between the two approaches (similar to the ImageNet experiments in Section 6). Again we notice that our method requires a lower learning rate for stability, and that switching parameterizations does not significantly impact accuracy.

## C. Source Code Release

An implementation of our technique can be found at `https://github.com/NREL/triangle_net`.

