# OpenReview forum: "Compelling ReLU Networks to Exhibit Exponentially Many Linear Regions at Initialization and During Training"
_ICML.cc/2025/Conference — ICML 2025 poster_

### Official Review · Reviewer_wXeQ · 2025-02-18

**Overall Recommendation:** 3

**Summary:**

This paper addresses the previously identified problem of bounded linear pieces in neural networks regardless of depth. The authors show that for a width 4 network, with a novel initialization and pertaining algorithm, they can maintain an exponential number of linear pieces with regard to the depth. They show that with this initialization and pretraining, followed by standard training, a toy net can achieve superior performances on a toy problem. They then discuss the extension to general networks, by dividing the general network into many toy networks and apply their method on each of them.

**Claims And Evidence:**

The claims made in the main text are clear, good, and interesting. In particular, these main claims are significant to solving the problem identified by previous works.

However, I found the claim made in Appendix A.11, which is arguably the most interesting one to the general audience, is a bit confusing.

**Essential References Not Discussed:**

I found the reference adequate.

**Experimental Designs Or Analyses:**

The experiments involving the toy dataset is good. However, I have a question regarding Table 1 and 2: why do the authors report minimum errors? I would say reporting the mean error ($L_2$ error) and max error ($L_\infty$ error) is more proper. After all, achieving a good minimum error sounds meaningless to the full picture.

I find the experiments regarding CIFAR and Imagenet, discussed in Appendix A.11, quite confusing due to the lack of sufficient details. First, the authors report the loss, but do not mention the loss type. I can hardly believe that with a standard cross entropy, such a small network could achieve loss less than 0.01 within 2 epochs. I suggest the authors replace Figure 16 with accuracies rather than loss values. Second, the authors do not describe the network they used in this experiment and how they apply their method as the activation in detail. Third, contrary to the experiments in the main body, this section only applies their method as part of the activation function. Why can't we follow the same design made in the main text? Does this still count as the same method? Fourth, they evaluate a single small network architecture in these experiments. Do they have a specific reason about not using a larger network on CIFAR-10? Could the benefits of their method vanish when we further increase the network size? An ablation on architecture helps.

I apologize for raising many questions about this appendix section. However, since this section presents the most relevant information about the practice, I do find it confusing and preliminary. In addition, this study deserves a place in the main text. While it is OK to not perform well on large datasets for this theoretical work, it is important to make it clear where and how large the gap is. Otherwise, follow-up works will be quite hard, which is also contrary to the authors' interest.

**Methods And Evaluation Criteria:**

The method has solid mathematical background and make sense. Evaluation is sufficient to demonstrate their claims made in the main text.

**Other Comments Or Suggestions:**

None.

**Other Strengths And Weaknesses:**

This work stands out in theoretical rigor and its sound evaluation on toy datasets. These evaluations support their construction quite well. However, as detailed above, when it comes to more relevant datasets, evaluations are flawed.

**Questions For Authors:**

See Section ''Experimental Designs Or Analyses''.

**Relation To Broader Scientific Literature:**

This work mainly solves the problem raised by [1].

[1] Hanin, B. and Rolnick, D. Deep ReLU networks have surprisingly few activation patterns.

**Theoretical Claims:**

Due to the large body of theoretical results and their complexity, I am unable to check their formal proof. However, the authors provided numerical demonstrations which agrees well with their theoretical results. Thus, I am inclined to believe their results.

---

> ### Author Rebuttal · Authors · 2025-04-01
>
> We thank the reviewer for their questions, and we appreciate that they took the time to read the appendix. The reason why the paper currently looks like it does, as Reviewer 1 (jsad) alludes to, is that the paper has been through several rounds of rebuttals, and so layers of preliminary experiments have built up over time and now the appendix is extensive. Originally, the authors’ main intention for the paper was to present the idea that better training algorithms could be created by finding efficient mathematical constructions that still permitted room for trainability. There are still many fundamental mathematical issues to be addressed to ‘properly’ set weights in higher dimensional settings, and this paper felt like it was at a natural stopping point even with only toy 1-d experiments. However, the reviewer makes a good point that it is valuable to know how far away the naive extension of this method to higher dimensions is from practical usage, and it likely makes sense to restructure the body of the paper to include both the one-dimensional experiments and the experiments from Section A.11. We will address this in our revised version.
>
> We will attempt to address the reviewer’s questions in order. In the one-dimensional experiments, the mean and minimum are not distances in function space, they’re the mean and minimum over 30 trials. The minimum is needed (and more important than the mean) because half the time the randomly initialized ReLU networks collapse from the dying ReLU issue. The minimum lets us see what happens when training ‘works’ for the various network types. We will clarify this in the manuscript.
>
> We agree that accuracy is probably a more interpretable and meaningful metric to display and can switch figure 16 accordingly.
>
> We do include a diagram of what using one of the small networks as an activation would look like at the beginning of the appendix. It could be beneficial, though, to describe the implementation further, especially if larger scale experiments are going to be moved to the main body. Essentially, each neuron in a dense layer gets its own copy of the 1-d experimental networks to learn some piecewise linear convex activation function for it to use. This adds layers to the network with a 4x4 block diagonal structure where each subnetwork’s version of that layer corresponds to one of the diagonal blocks. This can be implemented practically by turning the matrices into 4x4xW tensors (to remove all the 0 elements) and using pytorch’s einsum() function.
>
> The reason the method in the main body cannot apply directly to higher dimensional cases is that it only works for 1-d convex functions. Fortunately, activation functions are one-dimensional, and so that provides the most naive and easy route to scale up the construction. As we alluded to earlier, much more math is needed to figure out more elegant solutions.
>
> In light of your 4th question, it makes more sense to use a larger and better studied architecture such as VGG for the CIFAR-10 experiments, as it would make the results more compelling and less arbitrary. We can make that switch for the camera-ready version of the paper

---

> > ### Comment · Reviewer_wXeQ · 2025-04-02
> >
> > Dear authors,
> >
> > Thanks for the rebuttal. It clears most of my concerns except one:
> >
> > It is mentioned that half of random initializations collapse due to "dying ReLU". However, for Kaiming initialization etc., one specific advantage is to empirically mitigate the dying ReLU. As a result, when you init with Kaiming uniform, you hardly get training collapses (personally I never observed collapses from thousands of re-init across hundreds of datasets with Kaiming init). Could you provide more details on what collapse means for the proposed init (empirical details), and why the proposed init leads to this problem?

---

> > > ### Author Response · Authors · 2025-04-02
> > >
> > > As far as the authors’ understanding goes, kaiming initialization was designed to avoid exponential increase/decay in the signal magnitude by normalizing for the layer widths. This is a slightly different issue than the ‘dying relu’ phenomenon. The ‘dying relu’ issue happens if every neuron in a layer initializes such that its output is negative over the whole dataset. The ReLU activation then zeros out the entire layer, severing the network. In that case, the network output is constant. Wider layers make dying relu less likely to happen since more neurons gives the network more chances to get a signal through at each layer. Deeper networks increase the odds of dying relu because more layers gives dying relu more chances to happen. The reason you’ve likely never encountered this issue before is that most practical networks are operating with architectures that are much wider than they are deep; the networks we explore are 4 neurons wide with arbitrary depth, so they are somewhat exceptional in their shape.
> > >
> > > The dying relu problem can be lessened by the RAAI initialization, which was designed with the issue in mind, but since RAAI is still a random procedure it is still possible even if less likely. One of the advantages of our method is that each layer is being explicitly forced to compute triangles, the dying relu problem won’t happen. This is partly related to why in appendix A.10, the block diagonal constraints on random networks ruin their performance, but the higher dimensional version of our method is fine since block diagonality is the natural structure that arises from using the 1-d subnetworks as activations.
> > >
> > > We will clarify this in our revised manuscript, and we welcome any further discussion with the reviewer.  Thank you for your engagement.

---

### Official Review · Reviewer_ZVQa · 2025-03-06

**Overall Recommendation:** 2

**Summary:**

This work builds on the link between the expressiveness of a neural network with ReLU activation functions and its number of linear regions in the output space. The general idea is that the larger the number of linear regions in the output space, the higher the expressiveness of the network (i.e., the easier it would be for the network to implement complex functions).  Previous works show that the number of linear regions do not scale with depth for randomly initialized neural networks. The authors instead propose to initialize and (pre)train the model while constraining its weights such that the total network implements a chain of triangle functions, to make sure the number of linear regions scales exponentially with depth. They further illustrate through experiments that the proposed method achieves a higher accuracy than conventional models.

**Claims And Evidence:**

The actual claims made are dispersed throughout the paper and are somewhat vague. While I believe the claims might be true to a large extent, I think they are not sufficiently proven.

Referring to following lines in the paper:

*62, L: While an exponential increase in the expressiveness of a
ReLU network does not necessarily imply an exponential
increase in performance, one may intuitively expect a substantial
benefit, and our results bear this out*

*234, R: (1) we would like to determine how to learn the most effective
function representations possible, and (2) to explore
how the utilization of an increased number of linear regions
can affect a network’s ability to capture underlying
nonlinearity in its training data. To demonstrate that our
networks can learn function representations that better utilize
depth,*

*355, R; in particular,
compelling ReLU networks to approximate functions with
exponential accuracy as network depth is linearly increased.*


And assuming the following statements are correct:

-	randomly initialized networks have a number of linear regions that is not influenced by depth (the authors take this from literature),
-	for the proposed method, the number of linear regions exponentially increases with depth (this is true by design of the proposed method),

I recognize the following claims, according to my understanding of the paper / the above mentioned lines:
a)	the proposed method results in more linear regions w.r.t conventional models of the same depth,
b)	and, as a main claim: for the same depth, the proposed method results in higher accuracy w.r.t conventional models because of the increase in the number of linear regions (assuming this is the case, see claim a).

I think these claims are not sufficiently proven because:

Claim (a) -> In the experiments, there is no count of the number of linear regions for the conventional models. This potentially depends on the depth used in the experiments; it is thus unclear whether the proposed method really has more linear regions at all depths.

Claim (b) -> the authors make design choices that have effects similar to residual connections (line 218, R) and regularization (line 234, L). It is therefore unclear what the performance gains can be attributed to; i.e., if this is really because the number of linear regions is higher.  Some part of the performance gain could be due to the (implicit) use of residual connections/regularization when these are not present in the standard networks the authors compare against. Since there are no details on the architectures of these ‘standard’ networks, I cannot further verify this claim.

**Essential References Not Discussed:**

Not to my knowledge.

**Experimental Designs Or Analyses:**

See claims

**Methods And Evaluation Criteria:**

To prove that the proposed method leads to better accuracy, the used methods and evaluation seem sufficient for an exploratory paper. However, the methods do not sufficiently support the specific claims about why the method works better (as discussed above). Moreover, important details about the architecture of the standard models are missing.

**Other Comments Or Suggestions:**

This work is very interesting to me. But although it's also extensive, the "basis" of the paper in terms of claims and corresponding experiments is lacking. I would suggest revising this part. With an improved quality of the text and figures, I could be resubmitted.

**Other Strengths And Weaknesses:**

Strengths:
I think the proposed method is very interesting, and shows great promise from the preliminary experiments. The work done for the paper + supplemental seems substantial and could lead to an interesting contribution to the field.

Weaknesses:
-	the paper is confusing in its claims and goals. Is the goal to show that the proposed method ‘just works better’? That would need more extensive experiments with more details on the models the work compares against, plus an extension to real-world settings. Is the goal to prove that the method introduces more linear regions for larger depths, and this in turn leads to higher accuracy? Then the experiments should be changed to unequivocally prove this. While I think the work is very interesting, the “basis” of the paper is lacking.

-	While great care is taken at some points to explain the setup, the text is also often confusing and unclear. To give a specific example, starting at line 216, it is unclear what the actual (mathematical) goal is here. I’m guessing the authors wish to implement the triangle functions through making use of a neural network consisting of two nodes with ReLU activations, constraining the weights and biases such that the [0,1] input is mapped according to a triangle function with a peak at a. However, this is described in a very convoluted and unclear way in the text. It would be helpful to have clear mathematical descriptions, and more concise statements to prepare/guide the reader.

**Questions For Authors:**

/

**Relation To Broader Scientific Literature:**

The paper is well situated in the broader literature of expressiveness of ReLU networks, including other works that constrain and/or reparameterize the network to increase the expressiveness ( ElbrÅNachter et al. (2019), Chen & Ge (2024)). The origin and usefulness of triangle functions is also expenstively discussed. There is little to no literature referred that studies the relationship between the expressiveness of a network (in the sense of its potential to express certain functions) and the actual functions it can/does express after training with gradient descent (cfr. works on inductive bias) and/or the achieved accuracy.

**Theoretical Claims:**

There are some theoretical derivations in the appendix, which I did not check in detail. They do not support the main claims as I understood them (see above).

---

> ### Author Rebuttal · Authors · 2025-04-01
>
> We are glad to hear that the reviewer found the paper interesting and believes it shows promise.  We appreciate the reviewer’s willingness to reconsider their score in light of what we hope will clarify all the claims and goals of the paper. We agree that the extensiveness of the paper could cloud those, and we apologize for a lack of clarity - we hope we can provide a convincing preview here of how we plan to revise the paper to fix these writing issues.
>
> The goal of this paper is to present a method that forces ReLU networks to maintain exponentially many linear regions at initialization and during training, thus answering a five-year-old open question in the literature (Hanin and Rolnick, ICML 2019). We note that the goal is not to exponentially improve training speed, or similar goals - the general relationship between number of linear regions and training speed is unknown in the literature, and our paper does not attempt to address or solve this (though intuitively, more linear regions should usually help, and numerical results bear this out). The central claim of our paper (which is constructively proved) is that our triangle parameterization indeed achieves exponentially many linear regions, at least for a simple 1-d neural network case where we can rigorously analyze the effect of the triangle parameterization. We make some further, smaller claims about the importance (or not) of enforcing differentiability, and on extending the method to higher dimensions. Again, we hope this compactly summarizes what the paper is (and isn’t), and we will make this prominent (either text or bullet points) in the revised introduction of our paper. We will similarly revise the experiment descriptions and text to ensure their clarity for the camera-ready manuscript.
>
> In figures 5 and 7 the difference in the number of linear regions can be seen visually. In general, counting them exactly is combinatorially difficult (for example, appendix A.8 discusses Zaslavsky’s theorem - even for a shallow network, the number of regions grows to the power of the input dimension due to a relationship with Pascal’s triangle) and would likely be infeasible for many of the larger experiments.
>
> As mentioned in the response to reviewer 2 (a684), the triangle parameterization is not universal. The 1000-fold improvements are possible because the first phase of training can get the network into a neighborhood of parameter space where a solution that still uses all 2^n regions lives. But we would expect that with greater depths and demands for precision, that neighborhood size would shrink and this might not work for any curves that are not explicitly proven to be represented by weighted sums of composed triangle functions. Interestingly, this mirrors the developments of deep learning in general, where vanishing/exploding gradients and other such issues limited the depth of trainable models. Hopefully this work can serve as a starting point for constructing more expressive parameterizations.
>
> When we refer to networks as standard/ordinary/default/etc., we simply mean that there is nothing special about the network - it is a fully connected network of the same dimensions as our experimental models. This is still a fair comparison with our method even though some of the ‘jobs’ we give neurons might resemble residual connections or other empirical techniques because the standard networks have every opportunity to learn a residual neuron of their own, yet they do not. Our method comes entirely from theoretical constructions, so the fact that it bears similarity to empirical techniques ought to reinforce that both our method and the empirical techniques it is similar to are on the right track, rather than counting against us.
>
> In our experiments we do attempt to separate the effect of lots of regions from the mathematical regularization from Theorem 3.1 by training scaling parameters independently. We find that for 3 out of 4 convex functions it’s slightly worse when left unregularized, but still orders of magnitude better than random initialization.

---

> > ### Comment · Reviewer_ZVQa · 2025-04-07
> >
> > Dear authors,
> >
> > Thank you for your rebuttal. I have increased my score. However, while your results are interesting and promising, I still believe your manuscript is lacking in clarity. If you want to reach an audience beyond the researchers directly working on the same topic, I would advice to revise the manuscript such that the actual goal and method become more clear.

---

### Official Review · Reviewer_a684 · 2025-03-10

**Overall Recommendation:** 1

**Summary:**

This paper proposes an new parameterization and pretraining method for ReLU networks to ensure that the resulting function is a piecewise-linear mapping with the maximal number of "linear regions" (i.e., $2^{d}$ for a ReLU network of depth $d$). The motivation for such a parameterization appears to be the following: while the number of "linear regions" has been used in the literature as a proxy for expressive power, standard weight parameterizations (and associated randomized initialization schemes) for ReLU networks produce mappings with an average number of linear regions that is invariant to the depth $d$. The paper presents numerical experiments for learning 1-D and 2-D functions as well as training neural networks for image classification.

**Claims And Evidence:**

As far as I understand, the main claim is that "forcing" the number of linear regions to be exponential during the first stage of training can lead to significant improvements in approximation quality. Currently, the evidence is (in my opinion) quite limited:

- There are no "universal approximation" results for the proposed parameterization.
- The authors claim that their "pretraining algorithm acts as a preconditioner for, or guide to, the loss landscape". However there is no discussion of things like convergence rates, conditioning of the loss in the proposed parameterization etc.
- The numerical results on the largest-dimensional setting are limited to a small number of epochs and the description of the experiment is lacking (for example, the authors describe that they reduced the number of parameters "slightly" without specifying how). Other experiments yield worrying results: in Appendix A.10, the authors describe a dense and a block-diagonal version of a neural network model, and the new parameterization does not yield improvements in the dense case.

**Essential References Not Discussed:**

I would not consider the following a "classic" reference, but appears highly related to the structure of optimal weights for deep ReLU networks: https://stanford.edu/~pilanci/papers/geometric_algebra.pdf

**Experimental Designs Or Analyses:**

Please refer to my comments under "Claims and Evidence".

**Methods And Evaluation Criteria:**

Please refer to my comments under "Claims and Evidence".

**Other Comments Or Suggestions:**

N/A

**Other Strengths And Weaknesses:**

- The results on CIFAR-10 and Imagenet are promising (albeit for a very limited number of epochs; the exact experiment setup is also unclear). Consider moving them to the main text rather than hiding them in the appendix.
- In my opinion, the most interesting plot in this paper is the "learning rate stability" plot (Figure 13). The paper's main claim would be much stronger if the authors were able to produce such a plot for high-dimensional problems. Unfortunately, even in this plot, it is unclear if the learning rate is the same during the first and second stages of training.
- The presentation of the proposed parameterization needs to be improved. The extension to arbitrary dimensions, which is the most realistic setting, is not addressed at all in the main text.


I am overall interested in this paper and whether the claimed advantages persist in other problem settings, beyond simple regression and classification problems (e.g., solving inverse problems). However, in my opinion the paper needs considerable revision before it's ready for publication.

**Questions For Authors:**

N/A

**Relation To Broader Scientific Literature:**

The paper implicitly assumes that the number of linear regions is a good proxy for the approximation quality of the neural network. However, the literature includes work suggesting that overly expressive mappings are not necessarily good (without necessarily implying that deeper = worse). See, for example the following works: [1](https://arxiv.org/abs/2305.15598), [2](https://arxiv.org/abs/2209.15055), and references therein; see also [3](https://arxiv.org/abs/2011.04268) for a perspective on why regularization is important for solving inverse problems with deep neural networks.

**Theoretical Claims:**

I did not have time to verify the validity of Theorem 3.1.

---

> ### Author Rebuttal · Authors · 2025-04-01
>
> We appreciate the reviewer’s comments and perspective on our paper. In light of your observations, there are a few important aspects of this paper we would like to emphasize that were maybe not immediately clear.
>
> The first is regarding our claims about navigation of the loss landscape. When we say we’re improving navigation of the loss landscape, we’re not saying our paper is about proving convergence rates. Instead, what we mean is that we’ve simplified learning for the network so that it can discover better minima that it would not otherwise. In other words, we’ve disentangled the tasks of learning ‘how do I build functions efficiently?’ from ‘what does this data say?’ - the former question being one that ReLU + random initialization + gradient descent pathologically cannot answer.
>
> We agree that while having many linear pieces is necessary to approximate a nonlinear function, it isn’t always sufficient; those extra pieces could be useless or even detrimental if not regularized. We address this with Theorem 3.1 (Unfortunately due to length constraints, most of the math got separated from the main body). Theorem 3.1 is a mathematically principled and explicit method of regularization. It adapts the construction of x^2 presented in the introduction to work with asymmetric triangular waveforms, ensuring that the network output is differentiable by closing all the ‘gaps’ in the derivative (in the infinite depth limit). Without following Theorem 3.1, it’s possible to produce several different kinds of fractal structures that will likely not have good generalization properties.
>
> You mention that there are no universality results, and that is correct. The function family given by Theorem 3.1 is highly non-universal. x^2 is the only well-known member, and the other functions share a certain kind of dilated self-symmetry with x^2. However, even though it isn’t a universal parameterization, the triangle parameterization can get close enough to other one-dimensional convex curves that ordinary gradient descent can nudge the network to produce an otherwise unlearnably low loss (by a factor of up to 1000). This suggests that it’s potentially valuable to search for a better and more expressive parameterization to extend that result to more challenging settings; yet we believe these numerical results already merit sharing the present parameterization and strategy with the research community.
>
> You’ve also pointed out that the method when adapted to higher dimensions struggles outside of a block diagonal format. This result is actually to be expected, as all the mathematical development we have done is aimed at building 1-d convex functions using 4-neuron-wide networks. When the weight matrices are freed from the block diagonal constraint, the overwhelming majority of their weights need to be filled in with no additional mathematical insights, so it shouldn’t work well.
>
> The authors were pleasantly surprised that the experiments on real data work with any meaningful advantage, given the large mathematical gaps that remain unanswered. That is why the more practical results appear in the appendix (even though they are perhaps more interesting to the average ICML attendee). We see the value of this paper not as being a production ready method, or as being a complete mathematical theory of how to get exponentially better weight setting, but as containing important ideas that can help other researchers along in this direction. Replacing the bedrock algorithms of deep learning is probably a larger task than what one paper can accomplish, and this paper is at a natural stopping point where its substance requires a lot of space to convey, and the remaining challenges ahead are each nontrivial.
>
> We think the geometric algebra paper you’ve linked seems very interesting, and we’ll incorporate a citation to it. Also, we’ll work to clarify when the parameterization switch happens in the experiments.

---

### Official Review · Reviewer_JSad · 2025-03-19

**Overall Recommendation:** 4

**Summary:**

This paper shows how to build better regressions with ReLU feedforward networks. The key idea is to exploit the piecewise linear representation produced by such networks, with the philosophy that models with more of such pieces are likely to better interpolate the function of interest. These pieces are typically called linear regions in the ML literature. In the context of this paper, each piece corresponds to a different gradient of the function being approximated by the regression. For nonlinear functions, it stands to reason that exponentially many pieces are needed for ensuring a good overall approximation

There have been a number of papers that show how to obtain a model with an exponential number of linear regions on the network depth. In practice, however, the average number of linear regions is typically polynomial at best when the parameters are obtained from commonly used initializations and also from training by gradient descent regardless of how they were initialized.

The differentiation in this paper is that most of the training (described as pretraining) is carried out through a parameterized space of models of high expressiveness. By adjusting the peaks and valleys produced by each layer, the ordinary parameters (weights and biases) are obtained automatically from those choices. At the very end, regular training with Adam over the parameters is carried out for a limited time.

**In full disclosure, I have previously reviewed this paper at NeurIPS 2024 (reviewer 8qXf ). The authors addressed most of my questions then. I am surprised that this paper did not get in then, as it had only one opposing reviewer (scores were 7-7-5-3). **

**Claims And Evidence:**

The construction used by the authors for parameterizing models of high expressiveness is correct, and in fact known and explored by many authors before (Montufar et al., 2014; Telgarsky, 2015; Serra et al., 2018; Huchette et al., 2023) - all of which acknowledged in their work. I would argue that what they are doing is the next logical step: operationalizing trained neural networks with high expressiveness. The insight of doing that by parameterizing a subspace of models is their biggest contribution here. As they observe, this is an alternative to the use of splines, such as in KANs, to which I would add that preserving the model piecewise linear has algorithmic and computational advantages.

**Essential References Not Discussed:**

The authors do a good job with references (see "Claims and Evidence" above).

**Experimental Designs Or Analyses:**

I believe that the experiments are adequate. Unlike in the prior submission of this work, the authors have also explored nonconvex and bivariate functions. To be clear, in the context of mathematical optimization, piecewise linear approximations of nonlinear functions rarely go very far in terms of dimension. It may sound strange, but regression can be more challenging than classification.

I appreciate the effort of the authors in also including preliminary results about classification in the appendix, but I believe that this is not central to their work. There is plenty of work done in improving classification, for which reason even a meaningful contribution may have only a marginal impact when implemented. I believe that it is important the other reviewers understand this nuance.

**Methods And Evaluation Criteria:**

Yes.

**Other Comments Or Suggestions:**

Related to my comment about Theorem 3.1, I would appreciate if the authors were to write down the equations for each neuron. Figure 2 does a reasonable job, but treating the bias as a unit is confusing. Likewise, the discussion about the need for the sum unit is not entirely clear to me. I would have appreciated an example of before and after to explain the significance of having that unit.

In Figure 3, it would be helpful if the value of $a_i$ in each layer $i$ was given.

**Other Strengths And Weaknesses:**

See other items of this review.

**Questions For Authors:**

Please comment on my question about Theorem 3.1 above, and on explicitly writing the equation of each neuron.

If Figure 3, is the sum always 1 at x=1?

In Page 4, what do you mean by "would form a 1 -> 2 -> 1 -> 2 -> 1... shape"?

The terms "pretraining skipped", "differentiability not enforced", and "differentiability enforced" are not very clear. If I get it right, are these equivalent to "new initialization, no pretraining", "do pretraining by adjusting peaks and valleys without the sum unit", and "do pretraining by adjusting peaks and valleys with sum unit", all of which followed by regular training?

Why are you only reporting min values in Table 3, as opposed to min and mean as in Tables 1 and 2? That seems less informative.

**Relation To Broader Scientific Literature:**

See "Claims and Evidence" above.

**Theoretical Claims:**

Regarding Theorem 3.1 and surrounding discussion, I would appreciate if the authors precisely described the value of the network parameters in terms of the scaling factor $s_i$. If the zigzagging function being defined in each layer goes back and forth between 0 and 1, so that composing the function across layers produces an exponential number of pieces, then how can it work properly with scaling?

---

> ### Author Rebuttal · Authors · 2025-04-01
>
> We’re thrilled that you enjoyed our paper, and we appreciate your belief in the merits of this work. The sections you found confusing are things we agree we could clarify, so we’re extremely grateful for your guidance.
>
> In a single-hidden-layer network, each hidden neuron provides a basis function to the output (in that case the basis functions are ReLUs at different offsets and orientations). The networks in this paper are narrow and deep, and the basis we would like to use to build the output is the triangle waves produced at each layer (one peak, two peaks, 4 peaks, etc…). The sum neuron, acting similarly to a residual connection, is what allows these hidden features to pass through the remaining layers to be visible to the network output. Without the sum neurons, the network can only output a triangle wave with 2^d peaks. With the sum neurons, the network can perform the approximation of x^2 discussed in the introduction, as well as approximating a family of differentiable convex functions around it (which is the essence of theorem 3.1, it shows the correct coefficients for the sum neuron to use that will close the ‘holes’ in the derivative).
>
> Appendix A.1 gives the equations for the neurons in matrix form. It got separated from section 3 due to the length requirements. But we agree that a description of the individual neurons in equation form is probably more clear, so we will try to rework section 3 to do so. Perhaps the following is a more clear explanation of the network structure:
>
> Building a triangle:
> $t1(x) = ReLU(x)$ $t2(x) = ReLU(x-a)$ $output = \frac{t1}{a} - \frac{t2}{a-a^2}$
> The weight $\frac{1}{a}$ is chosen so that the output is 1 at $x=a$. the weight on $t2$ is equal to $\frac{1}{a} + \frac{1}{1-a}$ so that it negates $t1$ and then makes the output zero at $x=1$.
> This would give the diamond-shaped network shown in the top left of Figure 2. If we wanted to compose this network to make more oscillations, then the output node would become the input node for the next diamond-shaped network, which gives the awkward 1x2x1x2x1x2… pattern, which shows up in some of the background literature. Instead of collecting the $t1$ and $t2$ neurons into the output unit, they can be assembled directly in the input of the next layer using:
> $t1_{i+1} = ReLU (\frac{t1_i}{a_i} - \frac{t2_i}{a_i-a_i^2})$ $t2_{i+1} = ReLU (\frac{t1_i}{a_i} - \frac{t2_i}{a_i-a_i^2} - a_{i+1})$. (Maybe this is an unimportant distinction that can be skipped, and we could just give the equations in a constant-width format, or maybe it’s important to go over - we are open to feedback.)
>
> The sum neuron can be computed as:
> $sum_{i+1} = ReLU(sum_i - s_i*(\frac{t1_i}{a_i} + \frac{t2_i}{a_i-a_i^2}))$.
> The ReLU is irrelevant here since the output is always positive. We need to do one more trick to avoid having $s_i$ directly stored as a weight since it will be exponentially small (which could be problematic for storing or optimizing it). We use the network to iteratively apply the ratio $S_i = s_i/s_{i-1}$ in each layer, decaying the amplitude of the outputs of the t1 and t2 neurons. This also means that the bias has to be a neuron, so that it too can gradually scale down.
> $sum_{i+1} = ReLU(sum_i - S_i*(\frac{t1_i}{a_i} + \frac{t2_i}{a_i-a_i^2}))$ $t1_{i+1} = ReLU(S_i*(\frac{t1_i}{a_i} - \frac{t2_i}{a_i-a_i^2}))$ $t2_{i+1}) = ReLU(S_i*(\frac{t1_i}{a_i} - \frac{t2_i}{a_i-a_i^2} - a_{i+1}b_i))$ $b_{i+1} = ReLU(S_i*b_i)$
>
> In Figure 3 the sum will always be equal to 1 at $x=1$ since the network is set up to subtract the triangle waves from each layer from y=x, and the waves all output 0 at $x=1$.
>
> Labeling the tables is quite difficult since the experiments are complicated to describe, and there is only room for 2 or 3 words. You are correct about “Pretraining skipped” - which will encode the triangles into the network weights at initialization, but then only do ordinary gradient descent. The other two labels “differentiability enforced” and “differentiability not enforced” are about the scaling factors, rather than the sum neuron being present/absent. Theorem 3.1 is about how to pick the scaling factors ‘correctly’ to sum the waves in the sum neuron to get a differentiable output. Weighting the waves differently in the sum can give you fractals or other badly behaved functions in the output. The 1-dimensional experiments show that ‘holding gradient descent’s hand’ is actually helpful, and that choosing the scaling coefficients to make the output differentiable can act as an explicit regularizer.
>
> The reason only the minimum values are reported in table 3 is that all 4 of the functions are included, so we needed to save space. The minimum is more important to look at than the mean because we’re comparing against a standard random network, which will collapse from the dying relu issue half of the time (and thus have a disproportionately bad mean).

---

### Decision · Program_Chairs · 2025-05-01

**Decision:**

Accept (poster)

**Comment:**

This article provides a clever construction that allows one to force deep ReLU networks to compute functions with many more linear regions than is typically possible by a vanilla feed-forward network. While the practical utility of this is not clear, it is an interesting advance in the theory of ReLU networks and provides a thoughtful and fresh counterpoint to prior work in the subject.